# PFGuard: A Generative Framework with Privacy and Fairness Safeguards

**Soyeon Kim[1], Yuji Roh[2], Geon Heo[1], and Steven Euijong Whang[*1]**

[1] KAIST, {`purplehibird,geon.heo,swhang`}`@kaist.ac.kr`
[2] Google, `yujiroh@google.com`

## Abstract

Generative models must ensure both privacy and fairness for Trustworthy AI. While these goals have been pursued separately, recent studies propose to combine existing privacy and fairness techniques to achieve both goals. However, naïvely combining these techniques can be insufficient due to privacy-fairness conflicts, where a sample in a minority group may be represented in ways that support fairness, only to be suppressed for privacy. We demonstrate how these conflicts lead to adverse effects, such as privacy violations and unexpected fairness-utility tradeoffs. To mitigate these risks, we propose PFGuard, a generative framework with privacy and fairness safeguards, which simultaneously addresses privacy, fairness, and utility. By using an ensemble of multiple teacher models, PFGuard balances privacy-fairness conflicts between fair and private training stages and achieves high utility based on ensemble learning. Extensive experiments show that PFGuard successfully generates synthetic data on high-dimensional data while providing both DP guarantees and convergence in fair generative modeling.

## 1 Introduction

Recently, generative models have shown remarkable performance in various applications including vision (Wang et al., 2021b) and language tasks (Brown et al., 2020), while also raising significant ethical concerns. In particular, *privacy* and *fairness* concerns have emerged as generative models aim to mimic their training data. On the privacy side, specific training data can be memorized, allowing the leakage of personal sensitive information (Hilprecht et al., 2019; Sun et al., 2021). On the fairness side, any bias in the training data can be learned, resulting in biased synthetic data and unfair downstream performances across demographic groups (Zhao et al., 2018; Tan et al., 2020).

Although privacy and fairness are both essential for generative models, previous research has primarily tackled them separately. Differential Privacy (DP) techniques (Dwork et al., 2014), which provide rigorous privacy guarantees, have been developed for *private generative models* (Xie et al., 2018; Jordon et al., 2018); various fair training techniques, which remove data bias and generate more balanced synthetic data, have been proposed for *fair generative models* (Xu et al., 2018; Choi et al., 2020). To achieve both objectives, harnessing these techniques has emerged as a promising direction. For example, Xu et al. (2021) combine a fair pre-processing technique (Celis et al., 2020) with a private generative model (Chanyaswad et al., 2019) to train both fair and private generative models.

However, we contend that naïvely combining developed techniques for privacy and fairness can lead to a *worse privacy-fairness-utility tradeoff*, where utility is a model's ability to generate realistic synthetic data. We first illustrate how privacy and fairness can conflict in Fig. 1. Given the data samples $M_1$, $M_2$, $M_3$, and $m_1$ where $M$ and $m$ denote the majority and minority data groups, respectively, DP and fairness techniques play a tug-of-war re-

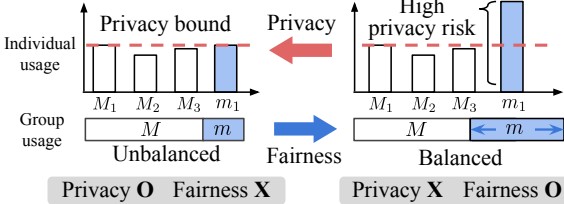

Figure 1: Privacy-fairness conflict. Privacy techniques prefer the left-hand scenario to prevent privacy risk of a certain data sample, while fairness techniques prefer the right-hand scenario to balance learning w.r.t. groups.

---

[*]Corresponding author

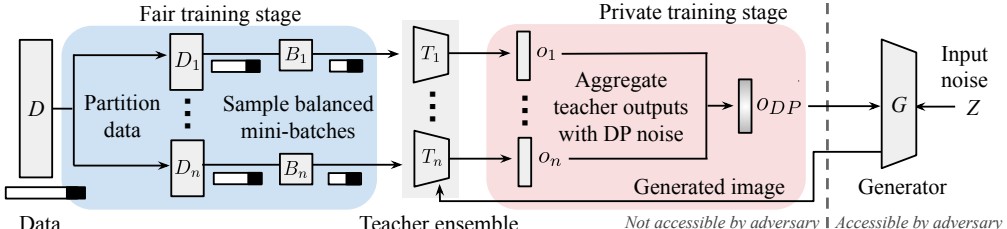

Figure 2: Overview of PFGuard. PFGuard integrates fairness and privacy into generative models through a two-stage process. In the fair training stage (blue), we train fair teacher models by sampling balanced mini-batches from biased data partitions. In the private training stage (red), we aggregate the teacher outputs with random noise to ensure Differential Privacy (DP). After training, only the trained DP generator is publicly released, safeguarding the privacy of all other components (e.g., the teacher ensemble). Through these training stages, PFGuard not only ensures fairness and privacy but also achieves high utility by leveraging ensemble learning of teacher models – resulting in unbiased, private, and high-quality synthetic data. More details on the framework are presented in Sec. 4.

garding the use of minority data point $m_1$; DP techniques *limit* its use to prevent privacy risks such as memorization, while fairness techniques *increase* its use to promote more balanced learning w.r.t. groups given the biased data. As a result, fairness techniques may undermine privacy by overusing $m_1$, while DP techniques may undermine fairness by limiting $m_1$'s usage. Moreover, combining different techniques can introduce new technical constraints, reducing the effectiveness of original methods. For instance, the fair preprocessing technique used by Xu et al. (2021) hinders the utility of the DP generative model by requiring data binarization, which incurs significant information loss on high-dimensional data such as images – restricting their overall framework only applicable to low-dimensional structural data.

Therefore, we design a generative framework that simultaneously addresses fairness and privacy while achieving utility for high-dimensional synthetic data generation. To this end, we propose PFGuard: a generative framework with **P**rivacy and **F**airness Safe**guard**s. As illustrated in Fig. 2, the key component is an *ensemble of intermediate teacher models*, which balances privacy-fairness conflicts between fair training and private training stages. In the *fair training* stage, we design a new sampling technique to train fair teachers, which provides a theoretical convergence guarantee to the fair generative modeling. In the *private training* stage, we employ the Private Teacher Ensemble Learning (PTEL) approach (Papernot et al., 2016; 2018), which aggregates each teacher's knowledge with random DP noise (e.g., noisy voting), to privatize the knowledge transfer to the generator. As a result, PFGuard provides a unified solution to train both fair and private generative models by transferring the teachers' fair knowledge in a privacy-preserving manner.

Compared to simple sequential approaches, PFGuard is carefully designed to address privacy-fairness conflicts. Recall that fairness techniques can incur privacy breaches by overusing minority data; in contrast, PFGuard *prevents privacy breaches* by decoupling fairness and privacy with intermediate teacher models. Although fair sampling can still compromise privacy in teacher models by potentially overusing minority data, PFGuard ensures privacy in the generator – our target model – by training it solely with the privatized teacher output, as shown in Fig. 2. Also, recall that privacy techniques can lead to fairness cancellation by suppressing the use of minority data; in contrast, PFGuard *avoids fairness cancellation* through teacher-level privacy bounding using PTEL approaches. Compared to sample-level privacy bounding methods like gradient clipping (Abadi et al., 2016), teacher-level bounding leaves room for teachers to effectively learn balanced knowledge via fair training. As a result, PFGuard provides strict DP guarantees for the generator and better preserves fairness compared to the combination of fairness-only and privacy-only techniques – see more analyses in Sec. 3.

Moreover, PFGuard is compatible with a wide range of existing private generative models and preserves their utility. PTEL is widely adopted in private generative models as it provides prominent privacy-utility tradeoff (Jordon et al., 2018; Chen et al., 2020; Long et al., 2021; Wang et al., 2021a). PFGuard can extend any of these models with a fair training stage as shown in Fig. 2, which requires a simple modification in the minibatch sampling process. Since additional fair sampling can be advantageous in maintaining optimization complexity compared to say adding a loss term for fairness, PFGuard preserves the privacy-utility tradeoff of PTEL as well while improving fairness. We also provide guidelines to control the fairness-privacy-utility tradeoff – see more details in Sec. 4.

Experiments in Sec. 5 show that PFGuard successfully generates high-dimensional synthetic data while ensuring both privacy and fairness; to our knowledge, PFGuard is the *first* framework that works on high-dimensional data including images. Our results also reveal two key findings: (1) existing private generative models can produce highly-biased synthetic data in real-world scenarios even with simple bias settings, and (2) a naïve combination of individual techniques may fail to achieve either privacy or fairness even with simple datasets – highlighting PFGuard's effectiveness and the need for a better integration of privacy and fairness in generative models.

**Summary of Contributions** **1)** We identify how privacy and fairness conflict with each other, which complicates the development of responsible generative models. **2)** We propose PFGuard, which is to our knowledge the first generative framework that supports privacy and fairness for high-dimensional data. **3)** Through extensive experiments, we show the value of integrated solutions to address the privacy-fairness-utility-tradeoff compared to simple combinations of individual techniques.

## 2 PRELIMINARIES

**Generative Models** We focus on Generative Adversarial Networks (GAN) (Goodfellow et al., 2014), which are widely-used generative models that leverage adversarial training of two networks to generate realistic synthetic data: 1) a generator that learns the underlying training data distribution and generates new samples and 2) a discriminator that distinguishes between real and generated data. The discriminator can be considered as the *teacher model* of the generator, as the generator does not have access to the real data and only learns from the discriminator via the GAN loss function.

**Differential Privacy** To privatize generative models, we use Differential Privacy (DP) (Dwork et al., 2014), a gold standard privacy framework that enables quantified privacy analyses. DP measures how much an adversary can infer about one data sample based on differences in two outputs of an algorithm $\mathcal{M}$, using two *adjacent datasets* that differ by one sample as defined below.

**Definition 2.1.** (($\varepsilon, \delta$)-Differential Privacy (Dwork et al., 2006a)) A randomized mechanism $\mathcal{M} : \mathcal{D} \to \mathcal{R}$ with range $\mathcal{R}$ satisfies ($\varepsilon, \delta$)-differential privacy if for any two adjacent datasets $\mathcal{D}, \mathcal{D}'$ and for any subset of outputs $\mathcal{O} \subseteq \mathcal{R}$, the following holds:

$$\Pr(\mathcal{M}(\mathcal{D}) \in \mathcal{O} \leq e^{\varepsilon} \Pr(\mathcal{M}(\mathcal{D}') \in \mathcal{O}) + \delta,$$

where $\varepsilon$ is the upper bound of privacy loss, and $\delta$ is the probability of breaching DP constraints.

We can enforce DP in an algorithm in two steps (Dwork et al., 2014). Given a target algorithm $f$ to enforce DP and a dataset $\mathcal{D}$, we first bound *sensitivity* (Def. 2.2), which captures the maximum influence of a single data sample on the output of $f$. We then add random noise with a scale proportional to the sensitivity value. A common way to add noise is to utilizing a Gaussian mechanism (Dwork et al., 2014) (Thm. 2.1), which uses Gaussian random noise with a scale proportional to $l_2$-sensitivity.

**Definition 2.2.** (Sensitivity (Dwork et al., 2014)) The $l_p$-sensitivity for a $d$-dimensional function $f : X \to \mathbb{R}^d$ is defined as $\Delta_f^p = \max_{\mathcal{D}, \mathcal{D}'} \|f(\mathcal{D}) - f(\mathcal{D}')\|_p$ over all adjacent datasets $\mathcal{D}, \mathcal{D}'$.

**Theorem 2.1.** (Gaussian mechanism (Dwork et al., 2014; Mironov, 2017)) Let $f : X \to \mathbb{R}^d$ be an arbitrary $d$-dimensional function with $l_2$-sensitivity $\Delta_f^2$. The Gaussian mechanism $\mathcal{M}_\sigma$, parameterized by $\sigma$, adds Gaussian noise into the output, i.e., $\mathcal{M}_\sigma(\boldsymbol{x}) = f(\boldsymbol{x}) + \mathcal{N}(0, \sigma^2 \boldsymbol{I})$, and satisfies ($\varepsilon, \delta$)-DP for $\sigma \geq \sqrt{2\ln(1.25/\delta)}\Delta_f^2/\varepsilon$.

**Fairness** We consider a generative model to be fair if two criteria are satisfied: 1) the model generates similar amounts of data for different demographic groups with similar quality, and 2) the generated data can be used to train a fair downstream model w.r.t. traditional group fairness measures. For 1), we measure the size and image quality disparities between the groups using the Fréchet Inception Distance (FID) score (Heusel et al., 2017; Choi et al., 2020) to assess image quality. For 2), we use two prominent group fairness measures: equalized odds (Hardt et al., 2016) where the groups should have the same label-wise accuracies; and demographic parity (Feldman et al., 2015) where the groups should have similar positive prediction rates.

## 3 PRIVACY-FAIRNESS CONFLICTS IN EXISTING TECHNIQUES

In this section, we examine the practical challenges of integrating privacy-only and fairness-only techniques to train both private and fair generative models. Based on Fig. 1's intuition on how

privacy and fairness conflict, we analyze how existing approaches for DP generative models and fair generative models can technically conflict with each other using standard DP and fairness techniques: DP-SGD (Abadi et al., 2016) and reweighting (Choi et al., 2020). Let $\boldsymbol{g}(\mathbf{x})$ denote the gradient of the data sample $\mathbf{x}$, and $p_{\text{bal}}$ and $p_{\text{bias}}$ denote balanced and biased data distributions, respectively.

- *DP-SGD* is a standard DP technique Chen et al. (2020) for converting non-DP algorithms to DP algorithms by modifying traditional stochastic gradient descent (SGD). Compared to SGD, DP-SGD 1) applies *gradient clipping* to limit the individual data point's contribution, where $\boldsymbol{g}(\mathbf{x})$ is clipped to $\boldsymbol{g}(\mathbf{x})/\max(1, \|\boldsymbol{g}(\mathbf{x})\|_2/C)$ (the sensitivity becomes the clipping threshold $C$), and 2) uses a *Gaussian mechanism* (Thm. 2.1) to add sufficient noise to ensure DP.

- *Reweighting* is a traditional fairness method (Horvitz & Thompson, 1952) widely used in generative modeling (Choi et al., 2020; Kim et al., 2024), which assigns *greater weights to minority groups* for a "balanced" loss during SGD. In particular, setting the sample weight to the likelihood ratio $h(\mathbf{x}_i)=p_{\text{bal}}(\mathbf{x}_i)/p_{\text{bias}}(\mathbf{x}_i)$ produces an unbiased estimate of $\mathbb{E}_{\mathbf{x}\sim p_{\text{bal}}}[\boldsymbol{g}(\mathbf{x})]$ as follows:

$$\mathbb{E}_{\mathbf{x}\sim p_{\text{bias}}}[\boldsymbol{g}(\mathbf{x})\cdot h(\mathbf{x})] = \mathbb{E}_{\mathbf{x}\sim p_{\text{bias}}}\Big[\boldsymbol{g}(\mathbf{x})\frac{p_{\text{bal}}(\mathbf{x})}{p_{\text{bias}}(\mathbf{x})}\Big] = \mathbb{E}_{\mathbf{x}\sim p_{\text{bal}}}[\boldsymbol{g}(\mathbf{x})]. \tag{1}$$

**Naïvely Adding Fairness Can Worsen Privacy**    Ensuring fairness in DP generative models can significantly increase *sensitivity* (Def. 2.2), leading to invalid DP guarantees. Sensitivity, which measures a data sample's maximum impact on an algorithm, is crucial in DP generative models because the noise amount required for DP guarantees is often determined by this sensitivity value. However, integrating fairness techniques in DP generative models can invalidate their sensitivity analyses by adjusting model outputs for fairness purposes. One example is the aforemetioned reweighting technique. To balance model training across groups, reweighting technique amplifies the impact of certain data samples (e.g., minority data samples), particularly by amplifying their gradients. However, performing reweighting on top of DP-SGD can incur privacy breaches, as *gradient amplified* beyond $C$ can cancel the *gradient clipping* in DP-SGD, invalidating the sensitivity $C$ provided from DP-SGD. Other examples include directly feeding data attributes such as class labels or sensitive attributes (e.g., race, gender) to a generator for more balanced synthetic data (Xu et al., 2018; Sattigeri et al., 2019; Yu et al., 2020), which can cause large fluctuations in the generator output and similarly end up increasing sensitivity. This increased sensitivity by fairness techniques requires more noise to maintain the same privacy level, compromising the original DP guarantees unless modifying DP techniques to add more noise. However, this modification is also not straightforward as assessing the increased sensitivity by fairness techniques can be challenging (Tran et al., 2021b).

**Naïvely Adding Privacy Can Worsen the Fairness-Utility Tradeoff**    Another direction is to ensure privacy in fair generative models, but configuring an *appropriate privacy bound* can be challenging, which can lead to unexpected and unstable fairness-utility tradeoffs.

In contrast to the previous demonstration, we can ensure privacy in reweighting-based fair generative models by replacing SGD to DP-SGD, guaranteeing a fixed sensitivity value $C$ via gradient clipping; however, finding $C$ that *balances* fairness and utility can be challenging. Gradient clipping now undoes the fairness adjustments, as reweighted gradients $\boldsymbol{g}(\mathbf{x})\cdot h(\mathbf{x})$ are clipped to $\boldsymbol{g}(\mathbf{x})\cdot h(\mathbf{x})/\max(1, \|\boldsymbol{g}(\mathbf{x})\cdot h(\mathbf{x})\|_2/C)$, and Eq. 1 does not hold if $C \leq \boldsymbol{g}(\mathbf{x})\cdot h(\mathbf{x})$. Here, one solution is to use a larger $C$ such that $C \geq \boldsymbol{g}(\mathbf{x})\cdot h(\mathbf{x})$. However, increasing $C$ also increases the noise required for DP, which reduces utility as illustrated in Fig. 3. Therefore,

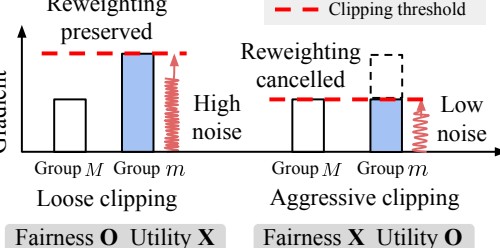

Figure 3: Fairness-utility tradeoff caused by DP-SGD when used on top of reweighting. Depending on the choice of $C$, DP-SGD may compromise utility (left) or fairness (right).

selecting a $C$ that balances fairness and utility may necessitate extensive hyperparameter tuning (Bu et al., 2024), complicating the systematic integration of DP into fair generative models.

Overall, we show that a naïve combination of existing fairness-only and privacy-only techniques can be insufficient to achieve both objectives. While we have not exhaustively covered all possible combinations, one can see how privacy breaches and unexpected fairness-utility tradeoffs can easily occur without a careful design. To avoid these downsides in naïve combinations, we emphasize the need for a *unified design* that integrates both privacy and fairness into generative models.

**Remark 1.** *We emphasize the need for a framework tailored to generative settings. There are notable fairness-privacy techniques for classification, but directly extending them to data generation can be challenging due to the fundamentally different goals of the two settings – see more details in Sec. A.*

## 4 FRAMEWORK

We now propose PFGuard, the first generative framework that simultaneously achieves statistical fairness and DP on high-dimensional data, such as images. As shown in Fig. 2, PFGuard balances privacy-fairness conflicts between fair and private training stages using an ensemble of teacher models as a key component. In Sec. 4.1, we first explain the *fair training stage*, which trains a fair teacher ensemble. In Sec. 4.2, we then explain the *private training stage*, which transfers the knowledge of this teacher ensemble to the generator with DP guarantees and ultimately trains a generator that is both fair and private. In Sec. 4.3, we lastly discuss how PFGuard's integrated design offers advantages in terms of fairness, utility, and privacy compared to the naïve approaches discussed in Sec. 3.

### 4.1 FAIR TRAINING WITH BALANCED MINIBATCH SAMPLING

**Intuition** We ensure fairness in the teachers by balancing the *minibatches* used for training. Here we assume a general training setup of stochastic gradient descent, where we iteratively pick a subset $\mathcal{B}$ of the training data (i.e., minibatches) to update model parameters more efficiently. Since generative models then only learn the underlying data distribution through $\mathcal{B}$, using *balanced mini-batches* $\mathcal{B} \sim p_{\text{bal}}$ will result in modeling $p_{\text{bal}}$ even if we have biased training data $\mathcal{D} \sim p_{\text{bias}}$. While another approach is to debias the training data itself by acquiring more minority data, this approach is costly and often infeasible for private domains with limited publicly-available data (Jordon et al., 2018).

**Theoretical Foundation** To develop a fair minibatch sampling technique with a convergence guarantee, we leverage *Sampling-Importance Resampling* (SIR) (Rubin, 1988; Smith & Gelfand, 1992) as the theoretical foundation. SIR is a statistical method for drawing random samples from a target distribution $\pi(x)$ by using a proposal distribution $\psi(x)$. SIR proceeds in two steps: 1) we draw a set of $n$ independent random samples $\mathcal{R}_1 = \{x_i\}_{i=1}^n$ from $\psi(x)$ and 2) we *resample* a smaller set of $m$ independent random samples $\mathcal{R}_2 = \{x_i\}_{i=1}^m$ from $\mathcal{R}_1$. Here, the resampling probability $w(x_i)$ is set proportional to $h(x_i) = \pi(x_i)/\psi(x_i)$, which is the likelihood ratio of a sample $x_i$ in $\pi(x)$ and $\psi(x)$. Then, the resulting samples in $\mathcal{R}_2$ are approximately distributed according to $\pi(x)$ as follows:

$$\Pr(x \leq t) = \sum_{i:x_i \leq t} w(x_i) = \sum_{i:x_i \leq t} \frac{h(x_i)}{\sum_i h(x_i)} = \frac{\sum_i \mathbb{1}\{x_i \leq t\}\pi(x_i)/\psi(x_i)}{\sum_i \pi(x_i)/\psi(x_i)} \qquad (2)$$

$$\xrightarrow[n \to \infty]{} \frac{\int \mathbb{1}\{x \leq t\}\{\pi(x)/\psi(x)\}\psi(x)dx}{\int \{\pi(x)/\psi(x)\}\psi(x)dx} = \int \mathbb{1}\{x \leq t\}\pi(x)dx \qquad (3)$$

where $\mathbb{1}(\cdot)$ is the indicator function. The distribution becomes exact when $n \to \infty$.

**Methodology** We now present our sampling technique, which samples $\mathcal{B} \sim p_{\text{bal}}$ based on SIR. We first make the following reasonable assumptions: 1) each data sample has a uniquely defined sensitive attribute $\mathbf{s} \in \mathcal{S}$ (e.g., race); 2) target $p_{\text{bal}}$ is a uniform distribution over $\mathbf{s}$; 3) following Choi et al. (2020), the same relevant input features are shared for each group $\mathbf{s}$ between the balanced and biased datasets (e.g., $p_{\text{bal}}(\mathbf{x}|\mathbf{s}{=}s) = p_{\text{bias}}(\mathbf{x}|\mathbf{s}{=}s)$), and similarly between the training dataset $\mathcal{D}$ and any subset $\mathcal{D}_i$ (e.g., $p_{\mathcal{D}}(\mathbf{x}|\mathbf{s}{=}s) = p_{\mathcal{D}_i}(\mathbf{x}|\mathbf{s}{=}s)$). We now outline the technique step-by-step below.

(1) We set the target distribution to $p_{\text{bal}}(\mathbf{x})$ and the proposal distribution to $p_{\text{bias}}(\mathbf{x})$, as our goal is to sample a balanced minibatch $\mathcal{B} \sim p_{\text{bal}}$ from the biased training dataset $\mathcal{D} \sim p_{\text{bias}}$.

(2) We divide $\mathcal{D}$ into $n_T$ disjoint subsets $\{\mathcal{D}_i\}_{i=1}^{n_T}$ to train teacher models $\{T_i\}_{i=1}^{n_T}$, such that each $\mathcal{D}_i$ is used for $T_i$ and retains the same distribution of $\mathbf{s}$ as $\mathcal{D}$ using the $\mathbf{s}$ labels (i.e., $p_{\mathcal{D}_i}(\mathbf{s}{=}s) = p_{\mathcal{D}}(\mathbf{s}{=}s)$). Then, we can derive $\mathcal{D}_i \sim p_{\text{bias}}$ using assumption 3) above as follows:

$$p_{\mathcal{D}_i}(\mathbf{x}) = \Sigma_s p_{\mathcal{D}_i}(\mathbf{x}|\mathbf{s}{=}s)p_{\mathcal{D}_i}(\mathbf{s}{=}s) = \Sigma_s p_{\mathcal{D}}(\mathbf{x}|\mathbf{s}{=}s)p_{\mathcal{D}}(\mathbf{s}{=}s) = p_{\text{bias}}(\mathbf{x}) \qquad (4)$$

(3) We sample $\mathcal{B}$ from $\mathcal{D}_i$ with a resampling probability $w(\mathbf{x})$ that is proportional to $h(\mathbf{x}) = p_{\text{bal}}(\mathbf{x})/p_{\text{bias}}(\mathbf{x})$, which is computed as follows:

$$h(\mathbf{x}) = \frac{p_{\text{bal}}(\mathbf{x})}{p_{\text{bias}}(\mathbf{x})} = \frac{p_{\text{bal}}(\mathbf{x}|\mathbf{s}{=}s)p_{\text{bal}}(\mathbf{s}{=}s)}{p_{\text{bias}}(\mathbf{x}|\mathbf{s}{=}s)p_{\text{bias}}(\mathbf{s}{=}s)} = \frac{p_{\text{bal}}(\mathbf{s}{=}s)}{p_{\text{bias}}(\mathbf{s}{=}s)} \simeq \frac{1/|\mathcal{S}|}{|\{\mathbf{x} \in \mathcal{D}|\mathbf{s}{=}s\}|/|\mathcal{D}|} \qquad (5)$$

where the second and third equality follows from assumption 1) and 3) above, respectively, and the last approximation follows from assumption 2) above and $\mathcal{D} \sim p_{\text{bias}}$.

A sample $\mathcal{B}$ from the above procedure is approximately distributed according to $p_{\text{bal}}$ based on SIR; we sample $\mathcal{D}_i$ from $p_{\text{bias}}$ and resample $\mathcal{B}$ from $\mathcal{D}_i$, where the resampling probability is proportional to $h(\mathbf{x}) = p_{\text{bal}}(\mathbf{x})/p_{\text{bias}}(\mathbf{x})$. Since a large number of minibatch samplings is needed to train generative models, the $\mathcal{B}$ distribution eventually converges to $p_{\text{bal}}$, leading to a fair generative modeling of $p_{\text{bal}}$.

**Extensions**   Our fair sampling technique is also extensible to private settings where the label of sensitive attribute $\mathbf{s}$ is *unavailable*, for example due to privacy regulations (Jagielski et al., 2019; Mozannar et al., 2020; Tran et al., 2022). In such settings, we can employ a binary classification approach to estimate $h(\mathbf{x})$ like Choi et al. (2020). While their work focuses on *non-private settings* and assumes an unbiased public reference data on the order of 10%–100% of $|\mathcal{D}|$ for the estimation, this assumption can be unrealistic in private domains due to the lack of public data. Our empirical study in Sec. 5.3 shows that we can achieve fairness with only 1–10% of the data, leveraging the ensemble learning of multiple-teacher structure, which can further reduce the estimation error. Note that in this extension, the convergence guarantee may not hold, as $\mathcal{D}_i \sim p_{\text{bias}}$ in step (2) might not be true in practice if the dataset is randomly partitioned without considering sensitive attribute labels.

## 4.2   Private Training with Private Teacher Ensemble Learning

**Intuition**   Although the fair sampling technique in Sec. 4.1 can enhance fairness, it can also introduce privacy risks due to inherent privacy-fairness conflicts (Sec. 3). Specifically, repeated resampling of certain data samples during teacher training can lead to memorization, compromising privacy in teacher models and ultimately affecting the generator during knowledge transfer. To address this privacy risk, we privatize *knowledge transfer* rather than the teacher models themselves, ensuring DP solely in the generator. Since the generator is only needed after training for synthetic data generation, we can only release DP generator to the public (Chen et al., 2020; Long et al., 2021).

**Privacy Guarantee**   We utilize Private Teacher Ensemble Learning (PTEL) (Papernot et al., 2016; 2018) to ensure DP during knowledge transfer. Unlike non-private ensemble learning, PTEL 1) trains each teacher model on a disjoint data subset and 2) adds noise proportional to the sensitivity of the aggregated knowledge (e.g., class labels (Papernot et al., 2018), gradients (Chen et al., 2020), etc.). Although aggregated knowledge can differ, sensitivity is commonly derived from *data disjointness*, where a single data point affects at most one teacher. For instance, GNMax aggregator (Papernot et al., 2018) aggregates predicted class label for a query input $\bar{\mathbf{x}}$ from teachers $\{T_i\}_{i=1}^{n_T}$ as follows:

$$\text{GNMax}(\bar{\mathbf{x}}) = \arg\max_{j}\{n_j(\bar{\mathbf{x}}) + \mathcal{N}(0, \sigma^2)\} \quad \text{for } j = 1, ..., c \tag{6}$$

where $n_j$ denotes the vote count for the $j$-th class (i.e., $n_j(\bar{\mathbf{x}}) = |\{i : T_i(\bar{\mathbf{x}})=j\}|$), and $\mathcal{N}(0, \sigma^2)$ denotes random Gaussian noise. Here, the $l_2$-sensitivity (Def. 2.2) is $\sqrt{2}$, as a single data point affects at most one teacher, increasing the vote counts by 1 for one class and decreasing the count by 1 for another class (see a more detailed analysis in Sec. B.2). Consequently, the GNMax aggregator satisfies $(\varepsilon, \delta)$-DP for $\sigma \geq \sqrt{8 \ln(1.25/\delta)}/\varepsilon$ based on the Gaussian mechanism (Thm. 2.1).

**Methodology**   PFGuard edits existing PTEL-based generative models to achieve both privacy and fairness by simply modifying the minibatch sampling process as described in Sec. 4.1. PTEL has been widely used to train DP generators (Jordon et al., 2018; Chen et al., 2020; Long et al., 2021; Wang et al., 2021a), resulting in various sensitivity analyses. While integrating fair training techniques can invalidate these analyses as discussed in Sec 3, PFGuard preserves any sensitivity value as long as PTEL enforce data disjointness; even with fair sampling, a single data point still affects only one teacher. PFGuard thus enhances fairness of various PTEL-based generative models, while preserving their own DP guarantees. We present formal DP guarantees and the training algorithm in Sec. C.

**Number of Teachers**   We provide guidelines on how to set the number of teachers $n_T$ for PFGuard, which affects the privacy-fairness-utility tradeoff. While $n_T$ is typically tuned via experiments (Long et al., 2021; Wang et al., 2021a), we set a maximum value of $n_T$ considering fairness. Since a large $n_T$ would result in a diverse ensemble that can generalize better, but also lead to a teacher receiving a data subset that is too small for training, we suggest $n_T$ to be at most $\lfloor |\mathcal{D}| \min_{s \in \mathcal{S}} p_{\text{bias}}(s) \rfloor$ where $\lfloor \cdot \rfloor$ denotes the floor function. Since this equation captures the size of the smallest minority data group,

this mathematical upper bound guarantees that each teacher *probabilistically* gets at least one sample of the smallest minority data group. In Sec. 5.3, we show how this bound helps avoid compromising fairness. We also discuss how to set $n_T$ when sensitive attribute labels are unavailable in Sec. C.3.

### 4.3 ADVANTAGES OF INTEGRATED DESIGN

We discuss how PFGuard overcomes the challenges of naïve approaches discussed in Sec. 3.

**Balances Privacy-Fairness Conflict**   PFGuard can sidestep privacy breaches and fairness cancellation arising from privacy-fairness conflicts. Compared to existing fairness-only techniques, which can compromise DP guarantees and require complex sensitivity assessments when applied to DP generators, PFGuard *automatically* preserves DP guarantees of PTEL-based DP generators by maintaining sensitivities, eliminating the need of complex assessments. Compared to existing privacy-only techniques, which often *directly* limit an individual sample's influence (e.g., gradient clipping discussed in Sec. 3) and can lead to fairness cancellation by suppressing the use of minority data, PFGuard uses *indirect* privacy bounding, in the sense that we limit the influence of *teacher models* in order to limit individual sample's influence. Since there are no DP constraints during the teacher learning, the teacher models can effectively learn balanced knowledge across data groups.

**Achieves Better Fairness-Utility Tradeoff**   The fair sampling of PFGuard introduces minimal training complexity, which can preserve utility while enhancing fairness. By modifying only the minibatch sampling process, PFGuard maintains the original loss function and avoids the additional fairness loss terms (Sattigeri et al., 2019; Yu et al., 2020) or auxiliary classifiers (Tan et al., 2020; Um & Suh, 2023) typically employed in fairness techniques. As a result, we show how PFGuard can achieve a prominent fairness-utility tradeoff with negligible computational overhead when integrated into PTEL-based generative models in Sec. 5. Further discussions on PFGuard's 1) flexibility with other fairness techniques and 2) synergy with adversarial learning are presented in Sec. C.3.

## 5 EXPERIMENTS

We perform experiments to evaluate PFGuard's effectiveness in terms of fairness, privacy, and utility.

**Datasets**   We evaluate PFGuard on three image datasets: 1) *MNIST* (LeCun et al., 1998) and *FashionMNIST* (Xiao et al., 2017) for various analyses and baseline comparisons, and 2) *CelebA* (Liu et al., 2015) to observe performance in real-world scenarios more closely related to privacy and fairness concerns. MNIST contains handwritten digit images, FashionMNIST contains clothing item images, and CelebA contains facial images. While MNIST and FashionMNIST are simplistic and less reflective of real-world biases, they enable reliable fairness analyses on top of high-performing DP generative models on these datasets, making them widely adopted in recent studies addressing the privacy-fairness intersections (Bagdasaryan et al., 2019; Farrand et al., 2020; Ganev et al., 2022). For CelebA, we resize the images to $32 \times 32 \times 3$ (i.e., *CelebA(S)*) and to $64 \times 64 \times 3$ (i.e., *CelebA(L)*) following the conventions in the DP generative model literature (Long et al., 2021; Wang et al., 2021a; Cao et al., 2021). We provide more dataset details (e.g., dataset sizes) in Sec. D.1.

**Bias Settings**   We create various bias settings across classes and subgroups, focusing on four scenarios: 1) *binary class bias*, which is a basic scenario often addressed in DP generative models, and 2) *multi-class bias*, *subgroup bias*, and *unknown subgroup bias*, which are more challenging scenarios typically addressed in fairness techniques, but not in DP generative models. We observe that DP generative models mostly perform poorly in these challenging scenarios, especially with complex datasets like CelebA, so we use MNIST for more reliable analyses. While recent privacy-fairness studies primarily focus on class bias in MNIST (Bagdasaryan et al., 2019; Farrand et al., 2020), we additionally analyze subgroup bias using image rotation for more fine-grained fairness analyses and to support prominent fairness metrics like equalized odds (Hardt et al., 2016). In all experiments, we denote $\mathbf{y} = 0$ as the minority class and $\mathbf{s} = 0$ as the minority group. More details on bias levels (e.g., size ratios between majority and minority data) and bias creation are in Sec. D.1.

**Metrics**   We evaluate utility, privacy, and fairness in both synthetic data and downstream tasks.
- *Utility*. We measure the overall and groupwise *Frechet Inception Distance* (FID) (Heusel et al., 2017) to evaluate the sample quality of synthetic data. We evaluate *model accuracy* in downstream tasks by training Multi-layer Perceptrons (MLP) and Convolutional Neural Networks (CNN) on synthetic data and testing on real datasets (Chen et al., 2020).

- *Fairness.* We measure the group size disparity in synthetic data with the *KL divergence* to uniform distribution $U(\mathbf{s})$ (i.e., $D_{KL}(p_G(\mathbf{s})||U(\mathbf{s}))$) (Yu et al., 2020) and the *distribution disparity* (i.e., $|p_G(\mathbf{s}) - U(\mathbf{s})|$) (Choi et al., 2020), where $p_G(\mathbf{s})$ denotes generated distribution w.r.t. $\mathbf{s}$. We measure the fairness disparities in downstream tasks as follows: *equalized odds disparity* (i.e., $\max_{y,s_1,s_2}|\Pr(\hat{\mathbf{y}}=y|Y=y, \mathbf{s}=s_1) - \Pr(\hat{\mathbf{y}}=y|\mathbf{y}=y, \mathbf{s}=s_2)|$, $\forall y \in \mathcal{Y}, s_1, s_2 \in \mathcal{S}$), *demographic disparity* (i.e., $\max_{s_1,s_2}|\Pr(\hat{\mathbf{y}}=1|\mathbf{s}=s_1) - \Pr(\hat{\mathbf{y}}=1|\mathbf{s}=s_2)|$, $\forall s_1, s_2 \in \mathcal{S}$), and *accuracy disparity* (i.e., $\max_{y_1,y_2}|\Pr(\hat{\mathbf{y}}=y_1|\mathbf{y}=y_1) - \Pr(\hat{\mathbf{y}}=y_2|\mathbf{y}=y_2)|$, $\forall y_1, y_2 \in \mathcal{Y}$).

- *Privacy.* We use privacy budget $\varepsilon$ for DP (Def. 2.1), which is preserved in both synthetic data and downstream tasks due to the post-processing property of DP (see Sec. B.1 for more details).

**Baselines** We compare PFGuard with three types of baselines: 1) privacy-only and fairness-only approaches for data generation, 2) simple combinations of these methods, and 3) recent privacy-fairness classification methods applicable to data generation. For 1) and 2), we use three state-of-the-art DP generative models – GS-WGAN (Chen et al., 2020), G-PATE (Long et al., 2021) and DataLens (Wang et al., 2021a) – and a widely-adopted fair reweighting method (Choi et al., 2020). For 3), we extend DP-SGD-F (Xu et al., 2020) and DPSGD-Global-Adapt (Esipova et al., 2022), which are fair variants of DP-SGD (Abadi et al., 2016). Specifically, we replace the DP-SGD used in GS-WGAN with these fairness-enhanced variants. We faithfully implement all baseline methods with their official codes and reported hyperparameters. More details on baseline methods are in Sec. D.2.

## 5.1 IMPROVING EXISTING PRIVACY-ONLY GENERATIVE MODELS

We evaluate how PFGuard enhances the performance of existing DP generative models. As PFGuard guarantees the same level of DP, we focus on the fairness and utility performances while fixing the privacy budget to $\varepsilon=10$, which is one of the most conventional values (Ghalebikesabi et al., 2023).

**Analysis on Synthetic Data** Table 1 shows the fairness and utility performances on synthetic data. Private generative models generally produce synthetic data with better overall image quality, but exhibit high group size disparity. In contrast, PFGuard significantly improves fairness by balancing group size and groupwise image quality, with a slight decrease in overall image quality.

Table 1: Fairness and utility performances of private generative models with and without PFGuard on synthetic data, evaluated on MNIST with subgroup bias under $\varepsilon=10$. Blue and red arrows indicate positive and negative changes, respectively. Lower values are better across all metrics.

| | Fairness | | Utility | | | | |
|---|---|---|---|---|---|---|---|
| Method | KL ($\downarrow$) | Dist. Disp. ($\downarrow$) | FID ($\downarrow$) | Y=1, S=1 | Y=1, S=0 | Y=0, S=1 | Y=0, S=0 |
| GS-WGAN | $0.177_{\pm 0.103}$ | $0.383_{\pm 0.097}$ | $\mathbf{77.97}_{\pm 2.25}$ | $\mathbf{95.58}_{\pm 3.35}$ | $155.20_{\pm 16.25}$ | $89.66_{\pm 0.79}$ | $101.39_{\pm 7.09}$ |
| G-PATE | $0.305_{\pm 0.011}$ | $0.522_{\pm 0.008}$ | $176.03_{\pm 3.03}$ | $182.50_{\pm 1.27}$ | $183.31_{\pm 2.99}$ | $178.89_{\pm 4.13}$ | $187.37_{\pm 3.51}$ |
| DataLens | $0.220_{\pm 0.030}$ | $0.450_{\pm 0.028}$ | $192.29_{\pm 3.67}$ | $197.13_{\pm 6.18}$ | $197.99_{\pm 6.01}$ | $202.86_{\pm 4.12}$ | $207.12_{\pm 12.75}$ |
| GS-WGAN + PFGuard | $\mathbf{0.067}_{\pm 0.036}$ ($\downarrow$) | $\mathbf{0.242}_{\pm 0.080}$ ($\downarrow$) | $83.67_{\pm 6.98}$ ($\uparrow$) | $114.54_{\pm 27.74}$ | $\mathbf{149.47}_{\pm 17.31}$ | $\mathbf{79.94}_{\pm 7.08}$ | $\mathbf{72.44}_{\pm 7.96}$ |
| G-PATE + PFGuard | $0.206_{\pm 0.062}$ ($\downarrow$) | $0.431_{\pm 0.066}$ ($\downarrow$) | $166.89_{\pm 21.61}$ ($\downarrow$) | $173.48_{\pm 19.93}$ | $173.79_{\pm 19.43}$ | $174.98_{\pm 24.06}$ | $185.92_{\pm 19.89}$ |
| DataLens + PFGuard | $0.161_{\pm 0.019}$ ($\downarrow$) | $0.389_{\pm 0.022}$ ($\downarrow$) | $200.23_{\pm 3.11}$ ($\uparrow$) | $209.74_{\pm 1.70}$ | $208.80_{\pm 0.39}$ | $207.03_{\pm 4.67}$ | $207.05_{\pm 3.17}$ |

**Analysis on Downstream Tasks** Table 2 shows the fairness and utility performances on downstream tasks. Compared to the synthetic data analysis, PFGuard enhances not only fairness, but also overall utility, especially for CNN models. We suspect that the increased overall utility results from the improved fairness in the input synthetic data, promoting more balanced learning among groups.

Table 2: Fairness and utility performances of private generative models with and without PFGuard on downstream tasks, evaluated on MNIST with subgroup bias under $\varepsilon=10$. Blue and red arrows indicate positive and negative changes, respectively.

| | MLP | | | CNN | | |
|---|---|---|---|---|---|---|
| | Fairness | | Utility | Fairness | | Utility |
| Method | EO Disp. ($\downarrow$) | Dem. Disp. ($\downarrow$) | Acc ($\uparrow$) | EO Disp. ($\downarrow$) | Dem. Disp. ($\downarrow$) | Acc ($\uparrow$) |
| GS-WGAN | $0.153_{\pm 0.030}$ | $0.061_{\pm 0.012}$ | $\mathbf{0.910}_{\pm 0.007}$ | $0.172_{\pm 0.045}$ | $0.069_{\pm 0.014}$ | $\mathbf{0.927}_{\pm 0.008}$ |
| G-PATE | $0.166_{\pm 0.082}$ | $0.063_{\pm 0.053}$ | $0.896_{\pm 0.005}$ | $0.256_{\pm 0.046}$ | $0.111_{\pm 0.001}$ | $0.888_{\pm 0.015}$ |
| DataLens | $0.226_{\pm 0.062}$ | $0.112_{\pm 0.035}$ | $0.867_{\pm 0.028}$ | $0.238_{\pm 0.044}$ | $0.110_{\pm 0.023}$ | $0.893_{\pm 0.022}$ |
| GS-WGAN + PFGuard | $\mathbf{0.067}_{\pm 0.029}$ ($\downarrow$) | $\mathbf{0.044}_{\pm 0.012}$ ($\downarrow$) | $0.900_{\pm 0.003}$ ($\downarrow$) | $\mathbf{0.063}_{\pm 0.059}$ ($\downarrow$) | $\mathbf{0.035}_{\pm 0.037}$ ($\downarrow$) | $\mathbf{0.927}_{\pm 0.009}$ ($-$) |
| G-PATE + PFGuard | $0.085_{\pm 0.052}$ ($\downarrow$) | $0.044_{\pm 0.033}$ ($\downarrow$) | $0.906_{\pm 0.008}$ ($\uparrow$) | $0.084_{\pm 0.036}$ ($\downarrow$) | $0.044_{\pm 0.011}$ ($\downarrow$) | $0.898_{\pm 0.023}$ ($\uparrow$) |
| DataLens + PFGuard | $0.169_{\pm 0.081}$ ($\downarrow$) | $0.106_{\pm 0.043}$ ($\downarrow$) | $0.859_{\pm 0.056}$ ($\downarrow$) | $0.141_{\pm 0.050}$ ($\downarrow$) | $0.078_{\pm 0.051}$ ($\downarrow$) | $0.898_{\pm 0.020}$ ($\uparrow$) |

Table 3: Comparison of privacy-fairness-utility performance on MNIST under $\varepsilon=10$, using GS-WGAN as the base DP generator (see Sec. D.2 for more details). The first three rows represent upper bound performances for vanilla, DP-only, and fair-only models. Evaluations cover both subgroup bias and unknown subgroup bias, where "no S" indicates whether the method is applicable without group labels. "perc" denotes the proportion of public data used compared to the training data size. "-" indicates no samples are generated. Lower values are better across all metrics, and we boldface the best results in each subgroup bias and unknown subgroup bias settings.

| | Privacy | Fairness | | | Utility | | | | |
|---|---|---|---|---|---|---|---|---|---|
| Method | $\varepsilon$ ($\downarrow$) | KL ($\downarrow$) | Dist. Disp. ($\downarrow$) | no S | FID ($\downarrow$) | Y=1, S=1 | Y=1, S=0 | Y=0, S=1 | Y=0, S=0 |
| Vanila | ✗ | 0.229 | 0.459 | ✗ | 31.95 | 28.01 | 55.53 | 44.04 | 63.50 |
| DP-only | 10 | 0.177 | 0.383 | ✗ | 77.97 | 95.58 | 155.20 | 89.66 | 101.39 |
| Fair-only | ✗ | 0.021 | 0.117 | ✓ | 38.62 | 50.78 | 52.69 | 75.46 | 53.86 |
| Reweighting | 13 | **0.009** | **0.044** | ✗ | 106.94 | 139.28 | 178.18 | 128.08 | 110.54 |
| DP-SGD → DP-SGD-F | 11 | 0.659 | 0.494 | ✗ | 90.20 | 121.78 | - | **73.07** | 159.83 |
| DP-SGD → DPSGD-GA | **10** | 0.693 | 0.707 | ✓ | 127.02 | 167.65 | - | 126.15 | - |
| **PFGuard** | **10** | 0.067 | 0.242 | ✗ | **83.67** | **114.54** | **149.47** | 79.94 | **72.24** |
| Reweighting (perc=1.0) | 13 | 0.025 | 0.148 | ✓ | 98.57 | 144.55 | 182.29 | 96.05 | 99.59 |
| Reweighting (perc=0.1) | 13 | 0.013 | 0.113 | ✓ | 106.94 | 139.28 | 178.18 | 128.08 | 110.54 |
| **PFGuard (perc=0.1)** | **10** | **0.004** | **0.041** | ✓ | **89.43** | **130.36** | **157.80** | **78.75** | **89.76** |

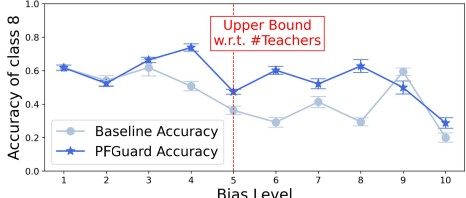

Figure 4: Fairness performances when varying bias levels ($\gamma$) given a fixed number of teachers, evaluated on MNIST with multi-class bias. We downsize the class '8' to $\gamma$ times smaller than the other classes to make it the minority class and use GS-WGAN as the baseline model.

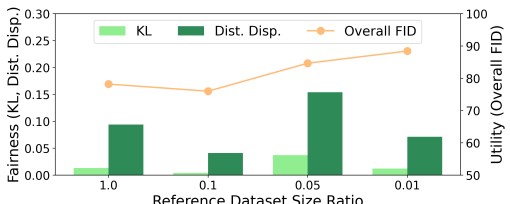

Figure 5: Fairness and utility performances for varying reference dataset size ratio compared to the training dataset size, evaluated on MNIST with unknown subgroup bias under $\varepsilon=10$. Lower values are better across all metrics used to evaluate fairness and utility.

## 5.2 PRIVACY-FAIRNESS-UTILITY TRADEOFF

We compare our privacy-fairness-utility performance against simple combinations of prior approaches. We evaluate performance under two bias settings: 1) subgroup bias and 2) unknown subgroup bias. Table 3 shows the results, which aligns with our privacy-fairness counteraction analysis in Sec. 3; fairness-only reweighting approaches compromise privacy due to the increased iterations, which may arise from modifying the loss function for fair training (i.e., the more a model uses the data, the weaker privacy it provides). In comparison, privacy-fairness classification techniques can maintain the original privacy guarantees, but significantly degrade utility and fairness, resulting in lower image quality and size disparities across groups – further discussions are provided in Sec. A. Among all methods, PFGuard is the only method that successfully 1) achieves both privacy and fairness and 2) preserves the closest utility to the original models. In Sec. E.1, we provide more analyses of the privacy-fairness-utility tradeoff, including Pareto Frontier results and varying privacy levels.

## 5.3 ABLATION STUDY

**Fairness Upper Bound on Number of Teachers**    We validate the proposed theoretical upper bound on the number of teachers for fairness, which depends on the bias level of the training data. To effectively simulate scenarios where a teacher receives only a small subset of minority data, we evaluate PFGuard in a multi-class bias setting, downsizing the minority class (i.e., class '8' for MNIST) by a factor of $\gamma$. Given that MNIST has fewer than 6,000 samples for class '8', our proposed upper bound is $\gamma \leq 5$ if we fix the number of teachers to 1,000. Fig. 4 shows that exceeding $\gamma=5$ leads to a noticeable accuracy drop for the minority class, which is consistent with our theoretical results. It is noteworthy that even with the decline, PFGuard shows higher accuracy than the privacy-only baseline, which shows a consistent decrease in accuracy for the minority as $\gamma$ increases.

**Impact of Reference Dataset Size**    We explore the influence of the reference dataset size when PFGuard is extended to unknown subgroup bias setting. Fig. 5 shows PFGuard achieves comparable fairness even with a small reference dataset size, while showing a slight increase in the overall FID.

**More Analyses** We provide more experiments in Sec. E, including a comparison of computation time (Sec. E.3), results on different datasets (Sec. E.2 and E.7), performance with additional normalization techniques (Sec. E.4), and a performance comparison with a different sampling strategy (Sec. E.5).

## 5.4    ANALYSIS WITH STRONGER PRIVACY, HIGH-DIMENSIONAL IMAGES

We provide preliminary results with CelebA dataset, which mirrors real-world scenarios with high-dimensional facial images. As our study is the first to address both privacy and fairness in image data, this exploration is crucial for understanding the challenges in real-world settings. To reflect the need of stronger privacy protection in practical applications, we limit the privacy budget to $\varepsilon=1$.

Table 4 shows the fairness and utility performances under these challenging conditions. We observe that DP generative models often exhibit extreme accuracy disparities even with a simplistic class bias setting, achieving over 90% accuracy for the majority class while achieving accuracy below 25% for the minority class. PFGuard consistently enhances the minority class performance and reduces accuracy disparity, while there is still room for improvement. Our results underscore the importance of tackling both privacy and fairness in future studies, encouraging more research in this critical area.

Table 4: Fairness and utility performances of private generative models with and without PFGuard on downstream tasks, evaluated on CelebA with binary class bias under $\varepsilon=1$. GS-WGAN is excluded due to lower image quality in this setting. Blue and red arrows indicate positive and negative changes, respectively. We provide the full results with standard deviations in Sec. E.8.

| | CelebA(S) | | | | CelebA(L) | | | |
|---|---|---|---|---|---|---|---|---|
| | Fairness | Utility | | | Fairness | Utility | | |
| Method | Acc. Disp. ($\downarrow$) | Acc ($\uparrow$) | Y=0 | Y=1 | Acc. Disp. ($\downarrow$) | Acc ($\uparrow$) | Y=0 | Y=1 |
| G-PATE | 0.978 | 0.666 | 0.014 | **0.992** | 0.968 | 0.668 | 0.023 | **0.991** |
| DataLens | 0.793 | 0.643 | 0.114 | 0.907 | 0.678 | 0.686 | 0.234 | 0.912 |
| G-PATE + PFGuard | 0.736 ($\downarrow$) | 0.678 ($\uparrow$) | 0.187 ($\uparrow$) | 0.923 ($\downarrow$) | **0.277** ($\downarrow$) | 0.563 ($\downarrow$) | **0.378** ($\uparrow$) | 0.655 ($\downarrow$) |
| DataLens + PFGuard | **0.725** ($\downarrow$) | **0.689** ($\uparrow$) | **0.205** ($\uparrow$) | 0.931 ($\uparrow$) | 0.641 ($\downarrow$) | **0.704** ($\uparrow$) | 0.276 ($\uparrow$) | 0.917 ($\uparrow$) |

## 6    RELATED WORK

We cover the private and fair data generation literature here and cover the 1) private-only data generation, 2) fair-only data generation, 3) privacy-fairness intersection literature in Sec. F. Compared to these lines of works, only a few works focus on private and fair data generation (Xu et al., 2021; Pujol et al., 2022). First, (Xu et al., 2021) proposes a two-step approach that 1) removes bias from the training data via a fair pre-processing technique (Celis et al., 2020) and 2) learns a DP generative model (Chanyaswad et al., 2019) from the debiased data. However, this framework is limited to low-dimensional structural data due to data binarization step in pre-precessing stage, which can incur significant information loss in high-dimensional image data. PFGuard, on the other hand, can generate high-dimensional image data with high quality. Second, (Pujol et al., 2022) proposes private data generation techniques satisfying causality-based fairness (Salimi et al., 2019), which consider the causal relationship between attributes. In comparison, PFGuard focuses on statistical fairness to achieve similar model performances for sensitive groups (Barocas et al., 2018). While causality-based approaches can better reveal the causes of discrimination than statistical approaches, modeling an underlying causal mechanism for real-world scenarios is also known to be challenging.

## 7    CONCLUSION

We proposed PFGuard, a fair and private generative model training framework. We first identified the counteractive nature between privacy preservation and fair training, demonstrating potential adverse effects – such as privacy breaches or fairness cancellation – when two objectives are addressed independently. We then designed PFGuard, which prevents the counteractions by using multiple teachers to harmonize fair sampling and private teacher ensemble learning. We showed how this integrated design of PFGuard offers multiple advantages, including a better fairness-privacy-utility tradeoff compared to other baselines, ease of deployment, and support for high-dimensional data.

ETHICS STATEMENT & LIMITATION

We believe our research addresses the critical issue of Trustworthy AI. Our focus on privacy and fairness underscores the need to design AI models that simultaneously safeguard individual privacy and mitigate biases in training data. In addition, our research and experiments are conducted with a strong commitment to ethical standards. All datasets used in this study, including publicly available human images, are widely used within the research community and do not contain sensitive or harmful content. There are also limitations, and we note that choosing the right privacy and fairness measures for an application can be challenging and also depends on the social context. We also note that the use of multiple teacher does increase the cost of training, but can provide more benefits particularly in balancing privacy and fairness.

REPRODUCIBILITY STATEMENT

All datasets, methodologies, and experimental setups used in our study are described in detail in the supplementary material. More specifically, we provide a description of the proposed algorithm in Sec. C.2, details of datasets and preprocessing in Sec. D.1, and implementation details including hyperparameters in Sec. D.2 to ensure reproducibility.

ACKNOWLEDGMENTS

This work was supported by Institute of Information & communications Technology Planning & Evaluation (IITP) grant funded by the Korea government (MSIT) (No. RS-2024-00509258, Global AI Frontier Lab). This work was supported by the National Research Foundation of Korea(NRF) grant funded by the Korea government(MSIT)(No. RS-2022-NR070121). This work was supported by the BK21 FOUR(Connected AI Education & Research Program for Industry and Society Innovation, KAIST EE, No. 4120200113769).

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

## A    CHALLENGES OF EXTENDING CLASSIFICATION TECHNIQUES

Continuing from Sec. 3, we provide more details of potential challenges when one tries to extend fairness-privacy classification techniques (Jagielski et al., 2019; Mozannar et al., 2020; Tran et al., 2021b; 2022) to generative settings due to the fundamentally different goals.

**Different DP Notions and Assumptions**    Classification and generation settings often concentrate on *different DP notions* or rely on *different assumptions*, which hinders simple extensions of techniques between them. In the classification setting, Differential Privacy *w.r.t. sensitive attribute* (Jagielski et al., 2019) is commonly addressed (Jagielski et al., 2019; Mozannar et al., 2020; Tran et al., 2021b; 2022), which considers the demographic group attribute as the only private information. This DP notion requires less DP noise compared to a more general notion of DP (Def. 2.1), which protects *all* input features, and enables a better privacy-utility tradeoff for DP classifiers. However, in the generative setting, a general notion of DP is mostly addressed, as presumed non-private features may in fact encode private information (e.g., pixel values in a facial image). Therefore, simply extending classification techniques to generative settings can be challenging, as it necessitates rigorous mathematical proofs for corresponding DP notions and may add a large DP noise when adapting to a general DP notion. Moreover, classification techniques can rely on convex objective functions (Tran et al., 2021a), but the assumption of convexity does not usually hold in generative models (Goodfellow et al., 2014).

**Challenges of Adjusting Privacy Bound**    While recent studies have proposed fair variants of DP-SGD (Xu et al., 2020; Esipova et al., 2022), directly adopting them in existing private generative models can undermine the original utility and privacy guarantee. To prevent aggressive gradient clipping in minority data groups, approaches to tune clipping threshold $C$ during training have been proposed, such as dynamically adjusting $C$ during training (Esipova et al., 2022) or utilizing different $C$ values w.r.t. groups (Xu et al., 2020). However, these adjustments of $C$ not only consume additional privacy budget, but also can significantly affect model utility, as private generative models often demonstrate high sensitivity in model convergence to these clipping values (Chen et al., 2020; Wang et al., 2021a; Dockhorn et al., 2022). Hence, given the limited privacy budget and the necessity to carefully set the value of $C$, these approaches of tuning $C$ may drastically change the original privacy-utility tradeoff of existing models to achieve fairness.

Here, we do not claim that extending classification techniques to generative settings is always impossible, but introduce the challenges that can complicate such extensions. To exemplify some possible cases, we extend classification methods from Esipova et al. (2022) and Xu et al. (2020) to generative settings, using them as baselines in our experiments – see Sec. 5 for results.

## B    DIFFERENTIAL PRIVACY

Continuing from Sec. 2 and Sec. 4.2, we provide more details on Differential Privacy (DP).

### B.1    POST-PROCESSING PROPERTY OF DIFFERENTIAL PRIVACY

Continuing from Sec. 2, we detail the post-processing property of DP as defined below.

**Theorem B.1.** (Post-processing (Dwork et al., 2014)) Let $\mathcal{M} : \mathcal{D} \to \mathcal{R}_1$ be a randomized mechanism that is $(\varepsilon, \delta)$-DP. Let $f : \mathcal{R}_1 \to \mathcal{R}_2$ be an arbitrary function. Then $f \circ \mathcal{M} : \mathcal{D} \to \mathcal{R}_2$ is $(\varepsilon, \delta)$-DP.

Due to the above post-processing theorem, a synthetic dataset $\tilde{\mathcal{D}} = G(\mathbf{z})$ generated from an $(\varepsilon, \delta)$-DP generator $G$ with random noise $\mathbf{z} \in \mathcal{Z}$ also satisfies $(\varepsilon, \delta)$-DP, as the random noise $\mathbf{z}$ is independent of the private training dataset used for the DP generator.

### B.2    SENSITIVITY ANALYSIS OF GNMAX AGGREGATOR

Continuing from Sec. 4.2, we echo the sensitivity analysis of GNMax aggregator provided by Papernot et al. (2018) for readers' convenience.

Given $\{T_i\}_{i=1}^{n_T}$ teachers, $c$ possible label classes, and a query input $\bar{\mathbf{x}}$, the teachers' vote counts for the $j$-th class to a query input $\bar{\mathbf{x}}$ is denoted as:

$$n_j(\bar{\mathbf{x}}) = |\{i : T_i(\bar{\mathbf{x}}) = j\}| \quad \text{for } j = 1, ..., c \tag{7}$$

where $T_i$ denotes the $i$-th teacher model. The vote count for each class is aggregated as follows:

$$\mathbf{n}(\bar{\mathbf{x}}) = (n_1, \ldots, n_c) \in \mathbb{N}^c \tag{8}$$

Since a single training data point only affects at most one teacher due to data disjointness, changing one data sample will at most change the votes by 1 for two classes, where we denote here as classes $i$ and $j$ without loss of generality. Given the two adjacent datasets $\mathcal{D}, \mathcal{D}'$ which differ by a single data point, let the aggregated vote counts be $\mathbf{n} = (n_1, \ldots, n_c)$ and $\mathbf{n}' = (n_1', \ldots, n_c')$, respectively. The $l_2$-sensitivity (Def. 2.2) can be derived as follows:

$$\Delta^2 = \max_{\mathcal{D}, \mathcal{D}'} \|(n_1, \ldots, n_c) - (n_1', \ldots, n_c')\|_2 \tag{9}$$

$$= \max_{n_i, n_i', n_j, n_j'} \|(0, \ldots, 0, n_i - n_i', 0, \ldots, 0, n_j - n_j', 0, \ldots, 0)\|_2 \tag{10}$$

$$= \max_{n_i, n_i', n_j, n_j'} \sqrt{(n_i - n_i')^2 + (n_j - n_j')^2} \leq \sqrt{2} \tag{11}$$

## C    PFGUARD FRAMEWORK

Continuing from Sec. 4.2, we provide more details on the proposed PFGuard framework.

### C.1    PRIVACY ANLAYSIS OF PFGUARD

Continuing from Sec. 4.2, we provide a theoretical proof of how PFGuard preserve the sensitivity of an arbitrary PTEL mechanism, and thus the sensitivity of an arbitrary PTEL-based generative model.

**PTEL-based Generative Models**    We first characterize the common training scheme of PTEL-based generative models (Jordon et al., 2018; Chen et al., 2020; Long et al., 2021; Wang et al., 2021a). To train a DP generator $G$ parametrized with $\theta_G$, the goal is to make its training algorithm $\mathcal{A} : \mathcal{D} \to G$ – which inputs the training dataset $\mathcal{D}$ and outputs the generator $G$ – satisfy DP (Long et al., 2021). Given the typical training scheme of stochastic gradient descent, which updates $\theta_G$ with gradient information $\boldsymbol{g}_G^{\text{up}}$, the training algorithm satisfies DP if $\boldsymbol{g}_G^{\text{up}}$ is privatized with a DP mechanism due to the post-processing property of DP (Thm. B.1) (Abadi et al., 2016). To produce privatized gradient $\tilde{\boldsymbol{g}}_G^{\text{up}}$, PTEL-based generative models typically proceed in two steps as follows:

(1) *Training teacher ensemble from disjoint data subsets.* The private dataset $\mathcal{D}$ is first divided into $n_T$ disjoint subsets $\{\mathcal{D}_i\}_{i=1}^{n_T}$, where each subset $\mathcal{D}_i$ is uniquely used to train a teacher model $T_i$ with its parameter $\theta_{T_i}$. Thus, a total of $n_T$ teachers are trained and form a teacher ensemble $T = \{T_i\}_{i=1}^{n_T}$.

(2) *Training generator by querying teacher ensemble.* The target generator $G$ with the parameter $\theta_G$ is trained by interacting solely with the teacher ensemble $T$, without accessing the original dataset $\mathcal{D}$. Given a random input noise $\mathbf{z}$, the generator $G$ generates an output $G(\mathbf{z})$ and queries the teacher ensemble $T$ with $G(\mathbf{z})$ for their supervision. Each teacher in $T$ votes on the gradient $\boldsymbol{g}_G^{\text{up}}$ for the query input $G(\mathbf{z})$, resulting in a vote count $\mathbf{n}(G(\mathbf{z}))$. This vote count is processed by the PTEL mechanism $\mathcal{M}$ (e.g., GNMax aggregator in Sec. 4.2) to finally produce a DP-sanitized gradient $\tilde{\boldsymbol{g}}_G^{\text{up}}$. The generator $G$ updates its parameter $\theta_G$ using $\tilde{\boldsymbol{g}}_G^{\text{up}}$.

Thus, given a PTEL mechanism $\mathcal{M}$ with $(\varepsilon, \delta)$-DP guarantee for each iteration and a total of $N$ training iterations, the total DP guarantee of the training algorithm $\mathcal{A}$ can be computed via the composition theorem of DP (Dwork et al., 2006b).

**Sensitivity Preservation**    We now provide a theoretical proof of how training with PFGuard preserves the original privacy analysis of an arbitrary PTEL-based generative model $G$ as long as data disjointness is enforced. As explained above, the privacy analysis of the PTEL-based generative model depends on the 1) $(\varepsilon, \delta)$-DP guarantee of the given PTEL mechanism $\mathcal{M}$ for each training iteration and the 2) total number of training iterations $N$. Here, we focus more on the *theoretical* DP guarantee than the *practical* DP guarantee, so we assume that $N$ is preserved with the fair sampling

algorithm $Sample(\cdot)$ used in PFGuard. That means, if $\mathcal{M}$ and $\mathcal{M} \circ Sample$ – PTEL mechanisms without and with PFGuard – result in the same $(\varepsilon, \delta)$-DP guarantee for each iteration, the same total DP guarantee is preserved. In particular, we show how $\mathcal{M}$ and $\mathcal{M} \circ Sample$ result in the same sensitivity (Def. 2.2), which ensures the same $(\varepsilon, \delta)$ values given the same amount of DP noise.

Since the PTEL mechanism $\mathcal{M}$ operates on a vote count $\mathbf{n}(\bar{\mathbf{x}})$ from the teacher ensemble $T$ given a generated sample as a query input (i.e., $\bar{\mathbf{x}} = G(\mathbf{z})$), we can denote $\mathbf{n}(\bar{\mathbf{x}}) = \mathbf{n}(\bar{\mathbf{x}}; T)$. Let $\mathcal{A}_T$ be the training algorithm of $T$, where $\mathcal{A}_T(\mathcal{D}_i) = T_i$ and $\mathcal{A}_T(\mathcal{D}) = T = \{T_i\}_{i=1}^{n_T}$ due to data disjointness used in training of $n_T$ teachers. Since $Sample(\cdot)$ – the proposed minibatch sampling algorithm – operates on $\mathcal{A}_T$ and samples a data subset $\mathcal{B}_i \subseteq \mathcal{D}_i$, data disjointness is preserved in $\mathcal{B}_i$. We thus can derive $\mathcal{A}_T(Sample(\mathcal{D}_i)) = \mathcal{A}_T(\mathcal{B}_i) = T_{s,i}$ and $\mathcal{A}_T(Sample(\mathcal{D})) = T_s = \{T_{s,i}\}_{i=1}^{n_T}$, resulting in a different teacher ensemble due to changed minibatches.

Let the original sensitivity value over $\mathbf{n}(\bar{\mathbf{x}}; T) = \mathbf{n}(\bar{\mathbf{x}}; \mathcal{A}_T(\mathcal{D}))$ be $k$. Then, due to the definition of the sensitivity, the following holds:

$$\Delta^2 = \max_{\mathcal{D}, \mathcal{D}'} \| \mathbf{n}(\bar{\mathbf{x}}; \mathcal{A}_T(\mathcal{D})) - \mathbf{n}(\bar{\mathbf{x}}; \mathcal{A}_T(\mathcal{D}')) \|_2 \tag{12}$$

$$= \max_{\mathcal{D}, \mathcal{D}'} \| \mathbf{n}(\bar{\mathbf{x}}; T_1, ..., T_{n_T-1}, T_{n_T}) - \mathbf{n}(\bar{\mathbf{x}}; T_1, ..., T_{n_T-1}, T'_{n_T})) \|_2 \tag{13}$$

$$= \max_{\mathcal{D}, \mathcal{D}'} \| (n_1, \ldots, n_c) - (n'_1, \ldots, n'_c) \|_2 \tag{14}$$

$$= \max_{n_i, n'_i, n_j, n'_j} \| (0, \ldots, 0, n_i - n'_i, 0, \ldots, 0, n_j - n'_j, 0, \ldots, 0) \|_2 \tag{15}$$

$$= \max_{n_i, n'_i, n_j, n'_j} \sqrt{(n_i - n'_i)^2 + (n_j - n'_j)^2} \leq k \tag{16}$$

where the second equality follows from 1) *adjacent datasets* $\mathcal{D}$ and $\mathcal{D}'$ differing by a single data sample and 2) *data disjointness*, where a single data sample can affect a particular teacher that receives the data partition including the data sample); we denote $T_{n_T}$ as the affected teacher without loss of generality. The last equality denotes how much one teacher can maximally contribute to the PTEL voting scheme, captured by the sensitivity value.

We now compute the sensitivity value over $\mathbf{n}(\bar{\mathbf{x}}; T_s) = \mathbf{n}(\bar{\mathbf{x}}; \mathcal{A}_T(Sample(\mathcal{D})))$ as follows:

$$\Delta^2 = \max_{\mathcal{D}, \mathcal{D}'} \| \mathbf{n}(\bar{\mathbf{x}}; \mathcal{A}_T(Sample(\mathcal{D}))) - \mathbf{n}(\bar{\mathbf{x}}; \mathcal{A}_T(Sample(\mathcal{D}'))) \|_2 \tag{17}$$

$$= \max_{\mathcal{D}, \mathcal{D}'} \| \mathbf{n}(\bar{\mathbf{x}}; T_{s,1}, ..., T_{s,n_T-1}, T_{s,n_T}) - \mathbf{n}(\bar{\mathbf{x}}; T_{s,1}, ..., T_{s,n_T-1}, T'_{s,n_T})) \|_2 \tag{18}$$

$$= \max_{\mathcal{D}, \mathcal{D}'} \| (v_1, \ldots, v_c) - (v'_1, \ldots, v'_c) \|_2 \tag{19}$$

$$= \max_{v_i, v'_i, v_j, v'_j} \| (0, \ldots, 0, v_i - v'_i, 0, \ldots, 0, v_j - v'_j, 0, \ldots, 0) \|_2 \tag{20}$$

$$= \max_{v_i, v'_i, v_j, v'_j} \sqrt{(v_i - v'_i)^2 + (v_j - v'_j)^2} \leq k \tag{21}$$

Here, the second equality follows similarly from adjacent datasets and data disjointness, where we denote $T_{s,n_T}$ as the affected teacher. Also, the last equality again captures how much one teacher can maximally contribute, which is a property of PTEL's voting scheme and is independent of the sampling algorithm – thus being k. As a result, using the sampling technique in PFGuard preserves the sensitivity value as long as the PTEL mechanism enforces data disjointness, although the sampling results in a different teacher ensemble compared to the original PTEL-based generative model. We note that this sensitivity analysis remains valid when $\mathcal{B}$ includes duplicate samples due to potential oversampling because it still affects one teacher due to data disjointness.

## C.2 TRAINING ALGORITHM

Continuing from Sec. 4.2, we provide the pseudocode describing the full training algorithm of PFGuard when integrated into a PTEL-based generative model.

As shown in Algorithm 1, PFGuard requires only a modification to the minibatch sampling process during teacher model training to achieve fairness (Line 5). Despite this modification, PFGuard

---

**Algorithm 1** Integrating PFGuard with PTEL-based generative models

---

**Input** Training dataset $\mathcal{D}$, ensemble of teacher model $T = \{T_i\}_{i=1}^{n_T}$ with each parameters $\theta_{T_i}$, batch size $B$, PTEL mechanism $\mathcal{M}(\mathbf{x}; T)$, teacher loss function $\mathcal{L}_T$, generator loss function $\mathcal{L}_G$
**Output** Differentially private generator $G$ with parameters $\theta_G$, total privacy cost $\varepsilon$
1:  Divide the dataset $\mathcal{D}$ into subsets $\{\mathcal{D}_i\}_{i=1}^{n_T}$
2:  **for** each training epoch **do**
3:      *///Phase 1: Fair Training*
4:      **for** each teacher model $T_i$ **do**
5:          Draw a minibatch $\{\mathbf{x}_i\}_{i=1}^{B} \subseteq \mathcal{D}_i$ with sampling ratio $w(\mathbf{x}) \propto h(\mathbf{x})$ using Eq. 5
6:          Draw a set of random noise $\{\mathbf{z}_i\}_{i=1}^{B}$ from input random noise distribution $p_z$ of $G$
7:          Update teacher model $T_i$ with $\mathcal{L}_{\mathcal{T}}(\theta_{T_i}; \mathbf{x}, G(\mathbf{z}; \theta_G))$
8:      **end for**
9:      *///Phase 2: Private Training*
10:     Draw a set of random noise $\{\mathbf{z}_i\}_{i=1}^{B}$ from input random noise distribution $p_z$ of $G$
11:     Generate synthetic data samples $G(\mathbf{z}; \theta_G)$
12:     Aggregate teacher output with PTEL mechanism $\tilde{o} \leftarrow \mathcal{M}(G(\mathbf{z}; \theta_G); T)$
13:         where the voting is on gradient of $\mathcal{L}_G(\theta_G)$
14:     Update generator model $G$ with $\tilde{o}$
15:     Accumulate privacy cost $\varepsilon$
16: **end for**
17: **return** Generator $G$, privacy cost $\varepsilon$

---

preserves the original privacy guarantee of the given PTEL-based generative model, as long as its sensitivity analysis relies on data disjointness – see Sec. C.1 for more details.

## C.3 EXTENSION

Continuing from Sec. 4.2, we provide more details on the extensibility of PFGuard.

**Setting Number of Teachers without Sensitive Attribute Labels**  We discuss how to extend the proposed upper bound on the number of teachers (i.e., $\lfloor |\mathcal{D}| \min_{s \in \mathcal{S}} p_{\text{bias}}(s) \rfloor$) when the sensitive attribute label $\mathbf{s}$ is unavailable. Note that the proposed bound does not rely on full knowledge of $p_{\text{bias}}(\mathbf{x})$ for data points to be generated (e.g., images), but instead relies on the distribution $p_{\text{bias}}(\mathbf{s})$ w.r.t. sensitive attributes. Thus, we can estimate the sensitive attribute distribution in the training data using traditional techniques such as K-means clustering (Macqueen, 1967) or random subset labeling (Forestier & Wemmert, 2016) and use the estimated distribution in the proposed bound. We note that while these estimations can be effective, they may introduce errors or additional overhead, such as increased computational time.

**Integration with Additional Fairness Techniques**  PFGuard supports additional integrations of existing fairness methods if they meet two conditions: 1) they apply exclusively to teacher models, ensuring no direct impact on the target generator, and 2) they maintain data disjointness – each sample affects only one teacher – which is a fundamental requirement for PTEL's privacy guarantees (see Sec. 4.2 for details). For example, GOLD (Mo et al., 2019), a fairness method designed to support Rawlsian Max-Min fairness (Joseph et al., 2016), is compatible with PFGuard. GOLD computes log density ratio estimates to identify worst-group samples and reweight the discriminator (i.e., teacher) loss to improve performance on these samples to achieve Rawlsian Max-Min fairness. Since this reweighting is applied solely to teacher models and preserves data disjointness, PFGuard with GOLD maintains the original privacy guarantees of the underlying PTEL generator – see privacy-fairness-utility tradeoff when additionally applying GOLD in Sec. E.6.

**Synergy with Adversarial Training**  While PFGuard's training scheme is loss-agnostic and does not necessarily require adversarial training, PFGuard can be particularly synergistic with adversarial training. We discuss with two reasons: (1) the min-max optimization and (2) the assumption of an optimal discriminator in adversarial training. First, adversarial learning optimizes conflicting goals (i.e., the min-max optimization) using the two components of the generator and the discriminator. Since privacy and fairness can also have conflicting goals, assigning them to separate components

may better balance these goals. Additionally, since the convergence of adversarial training depends heavily on the discriminator's optimality (Goodfellow et al., 2014), assigning fairness specifically to the discriminator may effectively guide the generator toward an optimal state that is also fair.

# D   EXPERIMENTAL SETTINGS

Continuing from Sec. 5, we provide more details on experimental settings. In all experiments, we use PyTorch and perform experiments using NVIDIA Quadro RTX 8000 GPUs. Also, we repeat all experiments 10 times and report the mean and standard deviation of the top 3 results. The reason we report the top-3 results is to favor the simple privacy-fairness baselines (e.g., "Reweighting" in Table 3), which we observe to fail frequently. We compare their best performances with PFGuard.

## D.1   DATASETS AND BIAS SETTINGS

Continuing from Sec. 5, we provide more details on datasets. We use three datasets: MNIST (LeCun et al., 1998), FashionMNIST (Xiao et al., 2017), and CelebA (Liu et al., 2015). *MNIST and FashionMNIST* contain grayscale images with 28 x 28 pixels and 10 classes. Both datasets have 60,000 training examples and 10,000 testing examples. *CelebA* contains 202,599 celebrity face images. We use the official preprocessed version with face alignment and follow the official training and testing partition (Liu et al., 2015). We mainly use image datasets instead of the traditional smaller tabular benchmarks for fairness because our goal is to make PFGuard work on higher dimensional data such as images. Additional results on tabular datasets are provided in Sec. E.2.

**MNIST & FashionMNIST**   We create four bias scenarios across classes and subgroups as follows.

- *Binary Class Bias*. For MNIST, we set digit "3" as the majority class $y = 1$ and "1" as the minority class $y = 0$. For FasionMNIST, we set "Sneakers" as $y = 1$ and "Trousers" as $y = 0$. For each class pair, we select two classes that share the fewest false negatives and thus can be considered independent, following the convention of prior approaches (Bagdasaryan et al., 2019; Farrand et al., 2020; Ganev et al., 2022). We set *bias level* as 2, which means the minority class $y = 0$ is 2 times smaller than the majority class $y = 1$. After creating bias, we apply random affine transformations to augment the datasets to match the original dataset size.

- *Multi-class Bias.* We set "8" as the minority class $y = 0$ and reduce its size while maintaining the size of the other 9 classes, following the aforementioned prior approaches. We vary the bias level from 1 to 10.

- *Subgroup Bias.* For both MNIST and FashionMNIST datasets, we use image rotation to define subgroups. We set non-rotated images as the majority group $s = 1$ and rotated image as the minority group $s = 0$. We also considered other options including adding lines and changing colors, but we observed that the other options often show the adverse affect of making the images noisier and thus reducing the model accuracy unnecessarily. The rotation also allows for simple and effective verification of subgroup labels in generated synthetic data by comparing the mean values of synthetic image vectors to the centroids of real image vectors. To validate this heuristic, we compared the results with 400 manually labeled images from each baseline model and observed high accuracy (e.g., 96.5% for MNIST).

- *Unknown Subgroup Bias.* In the previous subgroup bias setting, $s$ labels are not used during model training and only used for evaluation purposes after training.

**CelebA**   We create binary class bias using gender attributes, where we set female and male images as $y = 1$ and $y = 0$, respectively. As discussed in the main text, DP generative models often show low performance on CelebA in challenging bias scenarios like multi-class bias, which can hinder the reliability of fairness analyses (e.g., a random generator achieves perfect fairness by outputting random images regardless of data groups). Notably, we show that DP generative models can produce highly biased synthetic data even in this simple binary class bias setting (Table 4).

## D.2   BASELINES

Continuing from Sec. 5, we provide more details on baseline approaches used in our experiments.

**DP Generative Models**  We use three state-of-the-art PTEL-based generative models: GS-WGAN (Chen et al., 2020), G-PATE (Long et al., 2021), and DataLens (Wang et al., 2021a). For all models, we refer to their official Github codes to implement their models and to use their best-performing hyperparameters for MNIST, FashionMNIST, and CelebA.

- *GS-WGAN*. GS-WGAN is widely used in our experiments, as it leverages both PTEL and DP-SGD (Abadi et al., 2016) to ensure DP and thus allows for various integration with other techniques. Their approach first trains a multiple teacher ensemble and considers one teacher as the representative of the other teachers. Output of the representative teacher (i.e., gradients) is then sanitized with a DP-SGD based mechanism to train a DP generator. Compared to DP-SGD which operates on the whole minibatch, their DP mechanism operates on each data sample and thus can be considered as a composition of $B$ Gaussian mechanism where $B$ is the minibatch size. Our fair sampling preserves their sensitivity analyses despite the potential oversampling, as it does not change the sensitivity of each Gaussian mechanism on one input data (i.e., $2C$ due to the triangle inequality).

- *G-PATE and DataLens.* G-PATE and DataLens leverage teachers' votes on intermediate gradients to update the generator. To sanitize these teachers' votes with DP mechanisms, G-PATE uses a random projection and a gradient discretization, while DataLens uses a top-k stochastic sign quantization. Our fair sampling preserves their sensitivity analyses despite the potential oversampling, as each teacher still can throw only one vote.

**Privacy-Fairness Approaches**  We use 1) a prominent fair training approach based on reweighting (Choi et al., 2020) and 2) two recent classification techniques which address both privacy and fairness: DP-SGD-F (Xu et al., 2020) and DPSGD-Global-Adapt (Esipova et al., 2022).

- *Reweighting*. As outlined in Sec. 3, the reweighting approach computes the likelihood ratio and modifies the loss term of a discriminator to achieve fairness. For subgroup bias setting, we directly compute the likelihood ratio using sensitive group labels (Eq. 5). For unknown subgroup setting, we estimate the value using binary classification approach with their official Github code. During this estimation process, a public reference dataset is required to effectively train a binary classifier.

- *DP-SGD-F and DPSGD-Adapt-Global.* DP-SGD-F and DPSGD-Adapt-Global are fair variants of DP-SGD where clipping bounds are dynamically adjusted to control the fairness-utility tradeoff. To prevent excessive gradient clipping for minority data group samples, DP-SGD-F employs a groupwise clipping approach where each data group has its own clipping bound, while DPSGD-Adapt employs a scaling approach where all per-sample gradients are scaled down depending on a dynamically adjusted scaling factor. As DP-SGD-F and DPSGD-Adapt-Global do not provide the official codes to our knowledge, we faithfully implemented each algorithm based on their papers. We note that DP-SGD-F is not applicable in the unknown subgroup setting, as they require number of subgroup samples present in the batch to compute clipping bounds for each subgroup; in contrast, DPSGD-Global-Adapt is applicable in the unknown subgroup setting, as the scaling factor does not require knowledge on subgroup labels.

# E   ADDITIONAL EXPERIMENTS

Continuing from Sec. 5, 5.1, 5.3, and 5.4, we provide more experimental results.

## E.1   MORE EXPERIMENTS ON PRIVACY-FAIRNESS-UTILITY TRADEOFF

Continuing from Sec. 5.2, we provide more results on the privacy-fairness-utility tradeoff.

**Fairness-Utility Tradeoff with Varying Privacy Levels**  We analyze the fairness-utility tradeoff under varying privacy levels, following (Tran et al., 2021b). Specifically, we examine how training with PFGuard impacts the original fairness-utility tradeoff of underlying private generative models, using the MNIST dataset under the subgroup bias setting. The results are shown in Fig. 7 and Fig. 8. In weaker privacy regimes (i.e., higher values of $\varepsilon$), both the privacy-only generative model and PFGuard converge to similar performance levels. However, PFGuard show notable fairness improvements in stronger privacy regimes, albeit it also shows utility degradation due to slower

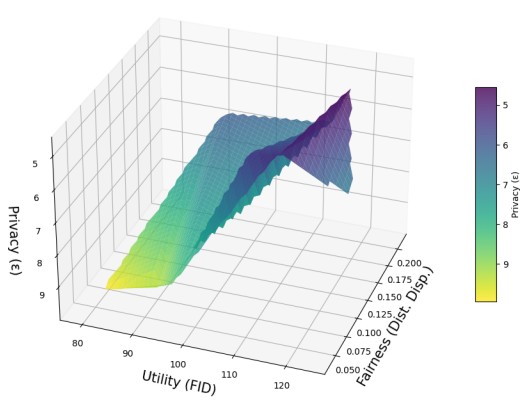

Figure 6: Visualization of Pareto frontier results of PF-Guard. A darker color indicates a more stronger privacy constraint. Both fairness metric (Distributon disparity) and utility metric (Overall FID) are lower the better.

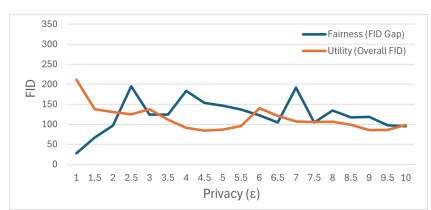

Figure 7: Fairness-utility tradeoff of GS-WGAN (i.e., private generative model).

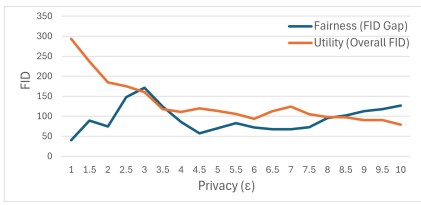

Figure 8: Fairness-utility tradeoff of GS-WGAN when trained with PFGuard.

convergence in the early training stages. We suspect this slower convergence is a natural consequence of learning a more balanced distribution, since the generative model should capture more diverse input features at the early training stage and thus can be more challenging.

**Pareto Frontier Results**  Fig. 6 visualizes the Pareto frontier results of PFGuard, showing the privacy-fairness-utility tradeoff. The nonlinear surface highlights the intricate relationship between these three objectives. As privacy constraints weaken (i.e., the light-colored region), both utility and fairness improve, converging toward a more favorable region. However, as privacy constraints become stronger, the utility narrows down to a specific range, while fairness greatly varies, which is particularly evident in the purple-colored region. This suggests that stronger DP noise consistently degrades the image quality, but results in highly variable fairness outcomes, which implies that there can be sweet-spot regions that can achieve both fairness and utility.

### E.2    EXPERIMENTAL RESULTS ON TABLUAR DATA

Continuing from Sec. 5.3, we provide additional results with a tabular dataset. Although tabular data are not of our immediate interest given our focus on scalability to high-dimensional data such as images, we show how PFGuard can support tabular data as well as image data.

**Experimental Setups**  *[Dataset]* We use Adult (Kohavi, 1996), which contains 43,131 examples of demographic information of individuals along with a binary label of whether their annual income is greater than 50k. We use gender as the sensitive attribute $\mathbf{s}$ and income as the class label $\mathbf{y}$. *[Baseline]* We use DP-WGAN (Xie et al., 2018) and PATE-GAN (Jordon et al., 2018) for private-only generative models, and FFPDG (Xu et al., 2021) for both fair and private generative models. Since PATE-GAN is a PTEL-based generative model, we use PATE-GAN as the base generator of PFGuard. We note that FFPDG is the most closest work of ours, which mainly focuses on tabular data and poses challenges when scaling to images – see more details in Sec. 6. *[Metric]* We use privacy and fairness metrics from the main text and AUROC as the utility metric, following FFPDG for effective comparison.

**Privacy-Fairness-Utility Tradeoff**  Table 5 shows the privacy-fairness-utility performances on the Adult dataset. Notably, PFGuard achieves fairness comparable to fair-only generative models while maintaining comparable utility with privacy-only generative models and FFPDG. In addition, Fig. 9 shows the Pareto frontier results compared to privacy-only generative models, where PFGuard outperforms other baselines by showing curves closer to the ideal region (i.e., low fairness discrepancy and high utility). These results on tabular data are consistent with the results on image data, highlighting the effectiveness and flexibility of PFGuard to a wide range of real-world applications by supporting both tabular and image datasets.

Table 5: Comparison of privacy-fairness-utility performance on Adult under $\varepsilon=1$, using PATE-GAN as the base DP generator. The first five rows represent upper bound performances for vanilla, DP-only, and fair-only models. Evaluations cover subgroup bias. Lower values are better across all metrics except AUROC. We boldface the best results and underline the second best results.

| | Privacy | Fairness | | Utility |
|---|---|---|---|---|
| Method | $\varepsilon$ | EO Disp. ($\downarrow$) | Dem. Disp. ($\downarrow$) | AUROC ($\uparrow$) |
| Vanila | ✗ | 0.56 | 0.58 | **0.80** |
| Fair-only | ✗ | **0.07** | **0.07** | 0.75 |
| DP-WGAN | ✓ | 0.31 | 0.30 | 0.69 |
| PATE-GAN | ✓ | 0.19 | 0.22 | 0.74 |
| RON-Gauss | ✓ | 0.18 | 0.14 | 0.70 |
| FFPDG | ✓ | 0.12 | 0.20 | 0.75 |
| **PFGuard** | ✓ | 0.08 | 0.12 | 0.76 |

Table 6: Comparison of computational time of private generative models with and without PFGuard.

| | MNIST | | | FashionMNIST | | |
|---|---|---|---|---|---|---|
| Method | w/o PFGuard | w/ PFGuard | Overhead (%) | w/o PFGuard | w/ PFGuard | Overhead (%) |
| GS-WGAN | 7378.30 | 7467.48 | 1.21 | 8114.96 | 8392.58 | 3.42 |
| G-PATE | 30810.25 | 31852.11 | 3.38 | 25238.84 | 26317.07 | 3.56 |
| DataLens | 41590.34 | 41638.19 | 0.12 | 547740.47 | 55714.41 | 1.78 |

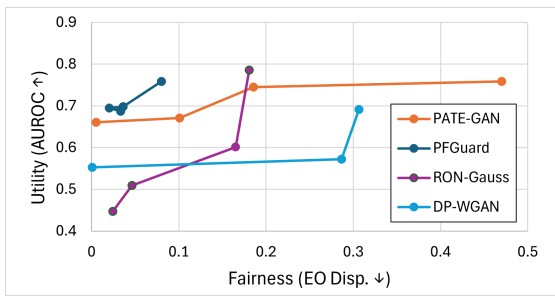

Figure 9: Pareto frontier results of DP generative models evaluated on tabular dataset (Adult). Upper left region denotes ideal case with low fairness discrepancy (EO Disparity) and high utility (AUROC).

Figure 10: FID values with different normalization factors, evaluated on MNIST using different batch sizes. GS-WGAN is used as the base DP generator. $N_1$ denotes a default normalization factor (Sec. E.4); $N_2$ denotes normalization factor from (Skare et al., 2003).

| | Normalization factor | |
|---|---|---|
| Batch size | $N_1$ | $N_2$ |
| 128 ($\varepsilon = 29.91$) | $75.05_{\pm2.26}$ | $\mathbf{74.58_{\pm2.96}}$ |
| 64 ($\varepsilon = 19.58$) | $75.35_{\pm6.67}$ | $\mathbf{72.20_{\pm4.58}}$ |
| 32 ($\varepsilon = 9.99$) | $82.68_{\pm7.13}$ | $\mathbf{78.18_{\pm1.85}}$ |

### E.3 COMPARISON OF COMPUTATIONAL TIME

Continuing from Sec. 5.3, we compare the computational time when integrating PFGuard with existing PTEL-based generative models. Table 6 shows that PFGuard incurs minimal overhead in computational time ($< 4\%$), due to the simple modification in minibatch sampling for fairness.

### E.4 ADDITIONAL NORMALIZATION TECHNIQUE FOR FASTER CONVERGENCE

Continuing from Sec. 5.3, we investigate the impact of the normalization factor on the overall image quality of PFGuard. While we use a traditional normalization factor $N_1 = \sum_i h(x_i)$ for $w(x_i) \propto h(x_i)$ in our SIR-based sampling algorithm, we can employ additional normalization techniques to boost the performance. For example, we can use $N_2 = \sum_i h(x_i)/N_{-i}$ for $w(x_i) \propto h(x_i)/N_{-i}$ where $N_{-i} = \sum_i h(x_i) - h(x_i)$, which is known to help faster convergence of SIR algorithms to the target distribution $p_{\text{bal}}$ (Skare et al., 2003). To investigate the influence on model convergence of normalization factors, we compare the overall image quality resulting from two different normalization factors $N_1$ and $N_2$ with varying batch sizes. We create both binary class bias and subgroup bias in the MNIST dataset with bias level 3 (see Sec. D.1 for details) to make a more challenging setting to effectively compare convergence speeds. We note that using a larger batch size can change the corresponding DP guarantee, as using large batches leads to more data usage.

Table 10 shows the comparison of image quality measured with FID, where a lower value indicates better quality. Although both $N_1$ and $N_2$ demonstrate comparable performance when using a large batch size, the performance gap becomes more evident as the batch size decreases. This empirical evidence shows that the performance of PFGuard can be further improved by additionally employing various normalization techniques.

### E.5    COMPARISON WITH A DIFFERENT SAMPLING STRATEGY

Continuing from Sec. 5.2, we provide a performance comparison with a different sampling scenario to enhance fairness. If sensitive attribute labels are available, one natural strategy is to train teachers exclusively on each data group to learn group representations, which is a commonly used strategy in stratified sampling approaches (Meng, 2013). We compare this stratified sampling strategy against PFGuard's importance sampling strategy, which trains teachers on randomly partitioned data, assessing their respective privacy-fairness-utility tradeoffs.

Table 7 shows that while stratified sampling achieves comparable fairness to importance sampling, it significantly reduces utility. A potential theoretical explanation is the downgraded utility of the teacher ensemble. Training teachers on non-i.i.d. datasets can lead to inconsistent convergence across teachers, leading to low consensus during teacher voting (Dodwadmath & Stich, 2022). This low consensus not only reduces the ensemble's prediction accuracy (i.e., utility), but also increases privacy costs during DP aggregation (Papernot et al., 2018) – ultimately leading to worse privacy-fairness-utility tradeoff. These results suggest that while training teachers on heterogeneous data may be useful in other contexts, importance sampling is better suited for ensemble learning context to achieve both privacy and fairness.

Table 7: Comparison of privacy-fairness-utility performance of sampling strategies, evaluated on MNIST under the subgroup bias setting. GS-WGAN is used as the base DP generator. Lower values are better across all metrics. We boldface the best results and underline the second best results.

|  | Privacy | Fairness | | Utility |
| --- | --- | --- | --- | --- |
| Method | $\varepsilon$ | KL ($\downarrow$) | Dist. Disp. ($\downarrow$) | FID ($\downarrow$) |
| DP-only | 10 | 0.177 | 0.383 | **77.97** |
| Stratified sampling | 10 | 0.090 | **0.209** | 135.35 |
| Importane sampling | 10 | **0.067** | 0.242 | 83.67 |

### E.6    EXTENSION WITH OTHER FAIRNESS TECHNIQUES

Continuing from Sec. 5.3 and Sec. C.3, we show experimental results on supporting other fairness techniques. In particular, we use GOLD (Mo et al., 2019), which aims to achieve Rawlsian Max-Min fairness (Joseph et al., 2016) and is compatible with PFGuard (see Sec. C.3 for more details).

Table 8 shows the privacy-fairness-utility performance when applying GOLD in the unknown sensitive attribute setting. GOLD achieves the best performance for the smallest (i.e., worst-case) group, aligning with its goal of Rawlsian Max-min fairness. However, GOLD does not surpass PFGuard in group fairness metrics or overall utility. These results suggest that incorporating additional fairness techniques can provide flexibility for different use cases, where GOLD can be particularly beneficial in applications prioritizing Rawlsian Max-min fairness over group fairness.

### E.7    EXPERIMENTAL RESULTS ON FASHIONMNIST

Continuing from Sec. 5.3, we show PFGuard's performances in synthetic data (Table 9) and downstream tasks (Table 10) evaluated on the FashionMNIST dataset. Compared to the results evaluated on MNIST, private generative models often generate more imbalanced synthetic data w.r.t. sensitive groups and exhibit lower overall image quality. In comparison, PFGuard consistently improves both fairness and overall utility in most cases, similar to the results observed in the MNIST evaluation.

Table 8: Comparison of privacy-fairness-utility performance of additional fairness technique (GOLD (Mo et al., 2019)), evaluated on MNIST under unknown subgroup bias setting. GS-WGAN is used as the base DP generator. Lower values are better across all metrics. We boldface the best results and underline the second best results.

|  | Privacy | Fairness | | | Utility |
|---|---|---|---|---|---|
| Method | $\varepsilon$ | KL ($\downarrow$) | Dist. Disp. ($\downarrow$) | Smallest group FID ($\downarrow$) | FID ($\downarrow$) |
| DP-only | 10 | 0.177 | 0.383 | 101.39 | **77.97** |
| PFGuard | 10 | **0.004** | **0.041** | 89.43 | 89.76 |
| PFGuard + GOLD | 10 | 0.090 | 0.209 | **84.52** | 100.39 |

Table 9: Fairness and utility performances of private generative models with and without PFGuard on synthetic data, evaluated on FashionMNIST with subgroup bias under $\varepsilon = 10$. Blue and red arrows indicate positive and negative changes, respectively. Lower values are better across all metrics.

|  | Fairness | | Utility | | | | |
|---|---|---|---|---|---|---|---|
| Method | KL ($\downarrow$) | Dist. Disp. ($\downarrow$) | FID ($\downarrow$) | Y=1, S=1 | Y=1, S=0 | Y=0, S=1 | Y=0, S=0 |
| GS-WGAN | $0.558_{\pm0.147}$ | $0.651_{\pm0.007}$ | $124.85_{\pm0.00}$ | $\mathbf{130.95}_{\pm0.00}$ | $278.06_{\pm0.00}$ | $155.36_{\pm0.00}$ | $217.00_{\pm0.00}$ |
| G-PATE | $0.270_{\pm0.026}$ | $0.494_{\pm0.021}$ | $245.13_{\pm24.85}$ | $271.28_{\pm15.32}$ | $249.66_{\pm10.95}$ | $275.61_{\pm38.26}$ | $282.58_{\pm30.95}$ |
| DataLens | $0.160_{\pm0.022}$ | $0.388_{\pm0.026}$ | $165.90_{\pm6.50}$ | $197.61_{\pm8.90}$ | $191.72_{\pm8.46}$ | $173.60_{\pm9.00}$ | $225.93_{\pm6.53}$ |
| GS-WGAN + PFGuard | $\mathbf{0.009}_{\pm0.065}$ ($\downarrow$) | $\mathbf{0.065}_{\pm0.049}$ ($\downarrow$) | $\mathbf{113.13}_{\pm7.24}$ ($\downarrow$) | $149.54_{\pm3.96}$ | $\mathbf{166.69}_{\pm10.00}$ | $114.87_{\pm12.26}$ | $146.67_{\pm22.52}$ |
| G-PATE + PFGuard | $0.190_{\pm0.050}$ ($\downarrow$) | $0.418_{\pm0.049}$ ($\downarrow$) | $242.20_{\pm42.70}$ ($\downarrow$) | $267.14_{\pm33.95}$ | $248.92_{\pm51.93}$ | $266.91_{\pm51.47}$ | $295.32_{\pm31.69}$ |
| DataLens + PFGuard | $0.127_{\pm0.037}$ ($\downarrow$) | $0.345_{\pm0.050}$ ($\downarrow$) | $209.48_{\pm12.01}$ ($\uparrow$) | $248.43_{\pm13.12}$ | $222.16_{\pm16.69}$ | $222.46_{\pm15.17}$ | $262.62_{\pm11.37}$ |

Table 10: Fairness and utility performances of private generative models with and without PFGuard on downstream tasks, evaluated on FashionMNIST with subgroup bias under $\varepsilon = 10$. Blue and red arrows indicate positive and negative changes, respectively.

|  | MLP | | | CNN | | |
|---|---|---|---|---|---|---|
|  | Fairness | | Utility | Fairness | | Utility |
| Method | EO Disp. ($\downarrow$) | Dem. Disp. ($\downarrow$) | Acc ($\uparrow$) | EO Disp. ($\downarrow$) | Dem. Disp. ($\downarrow$) | Acc ($\uparrow$) |
| GS-WGAN | $0.773_{\pm0.019}$ | $\mathbf{0.021}_{\pm0.019}$ | $0.812_{\pm0.009}$ | $0.795_{\pm0.008}$ | $\mathbf{0.007}_{\pm0.007}$ | $0.804_{\pm0.003}$ |
| G-PATE | $0.636_{\pm0.065}$ | $0.162_{\pm0.059}$ | $0.875_{\pm0.004}$ | $0.525_{\pm0.056}$ | $0.095_{\pm0.064}$ | $0.884_{\pm0.010}$ |
| DataLens | $0.484_{\pm0.168}$ | $0.203_{\pm0.092}$ | $\mathbf{0.901}_{\pm0.030}$ | $\mathbf{0.328}_{\pm0.039}$ | $0.072_{\pm0.045}$ | $\mathbf{0.925}_{\pm0.009}$ |
| GS-WGAN + PFGuard | $\mathbf{0.296}_{\pm0.099}$ ($\downarrow$) | $0.152_{\pm0.033}$ ($\uparrow$) | $0.884_{\pm0.015}$ ($\uparrow$) | $0.449_{\pm0.082}$ ($\downarrow$) | $0.203_{\pm0.037}$ ($\uparrow$) | $0.910_{\pm0.011}$ ($\uparrow$) |
| G-PATE + PFGuard | $0.556_{\pm0.152}$ ($\downarrow$) | $0.154_{\pm0.087}$ ($\downarrow$) | $0.885_{\pm0.013}$ ($\uparrow$) | $0.476_{\pm0.051}$ ($\downarrow$) | $0.124_{\pm0.041}$ ($\uparrow$) | $0.899_{\pm0.017}$ ($\uparrow$) |
| DataLens + PFGuard | $0.387_{\pm0.154}$ ($\downarrow$) | $0.153_{\pm0.103}$ ($\downarrow$) | $0.858_{\pm0.025}$ ($\downarrow$) | $0.394_{\pm0.109}$ ($\uparrow$) | $0.093_{\pm0.074}$ ($\uparrow$) | $0.877_{\pm0.025}$ ($\downarrow$) |

## E.8 Full Results with Standard Deviation

Continuing from Sec. 5.2 and Sec. 5.4, we show full results with standard deviations. Table 11 and Table 12 show the full results of Table 4, demonstrating the stable fairness improvements of PFGuard while preserving utility.

Table 11: Full results of fairness and utility performances of private generative models with and without PFGuard on downstream tasks, evaluated on CelebA(S) with binary class bias under $\varepsilon = 1$. GS-WGAN is excluded due to lower image quality in this setting. Blue and red arrows indicate positive and negative changes, respectively.

|  | Fairness | Utility | | |
|---|---|---|---|---|
| Method | Acc. Disp. ($\downarrow$) | Acc ($\uparrow$) | Acc (Y=1) | Acc (Y=0) |
| G-PATE | $0.978_{\pm0.024}$ | $0.666_{\pm0.003}$ | $0.014_{\pm0.014}$ | $\mathbf{0.992}_{\pm0.010}$ |
| DataLens | $0.793_{\pm0.173}$ | $0.643_{\pm0.031}$ | $0.114_{\pm0.087}$ | $0.907_{\pm0.087}$ |
| G-PATE + PFGuard | $0.736_{\pm0.126}$ ($\downarrow$) | $0.678_{\pm0.003}$ ($\uparrow$) | $0.187_{\pm0.085}$ ($\uparrow$) | $0.923_{\pm0.041}$ ($\downarrow$) |
| DataLens + PFGuard | $\mathbf{0.725}_{\pm0.055}$ ($\downarrow$) | $\mathbf{0.689}_{\pm0.004}$ ($\uparrow$) | $\mathbf{0.205}_{\pm0.040}$ ($\uparrow$) | $0.931_{\pm0.015}$ ($\uparrow$) |

Table 12: Full results of fairness and utility performances of private generative models with and without PFGuard on downstream tasks, evaluated on CelebA(L) with binary class bias under $\varepsilon = 1$. GS-WGAN is excluded due to lower image quality in this setting. Blue and red arrows indicate positive and negative changes, respectively.

| | Fairness | Utility | | |
|---|---|---|---|---|
| Method | Acc. Disp. ($\downarrow$) | Acc ($\uparrow$) | Acc (Y=1) | Acc (Y=0) |
| G-PATE | $0.968_{\pm 0.025}$ | $0.668_{\pm 0.001}$ | $0.023_{\pm 0.018}$ | $\mathbf{0.991_{\pm 0.007}}$ |
| DataLens | $0.678_{\pm 0.027}$ | $0.686_{\pm 0.011}$ | $0.234_{\pm 0.028}$ | $0.912_{\pm 0.005}$ |
| G-PATE + PFGuard | $\mathbf{0.277_{\pm 0.314}}$ ($\downarrow$) | $0.563_{\pm 0.028}$ ($\downarrow$) | $\mathbf{0.378_{\pm 0.181}}$ ($\uparrow$) | $0.655_{\pm 0.133}$ ($\downarrow$) |
| DataLens + PFGuard | $0.641_{\pm 0.038}$ ($\downarrow$) | $\mathbf{0.704_{\pm 0.007}}$ ($\uparrow$) | $0.276_{\pm 0.031}$ ($\uparrow$) | $0.917_{\pm 0.008}$ ($\uparrow$) |

## F  RELATED WORK

Continuing from Sec. 6, we provide more discussion on related work.

**Private-only Data Generation**  Most privacy-preserving data generation techniques focus on satisfying differential privacy (DP) (Dwork et al., 2014). The majority of these techniques focus on privatizing Generative Adversarial Networks (GANs) (Goodfellow et al., 2014), while recent studies have explored other generative models as well (Takagi et al., 2021; Cao et al., 2021; Harder et al., 2021; Liew et al., 2021; Chen et al., 2022; Vinaroz et al., 2022; Yang et al., 2023; Ghalebikesabi et al., 2023). One privatization approach is based on DP-SGD (Abadi et al., 2016), which is a DP version of the standard stochastic gradient descent algorithm to train ML models (Xie et al., 2018; Zhang et al., 2018; Torkzadehmahani et al., 2019; Bie et al., 2023). Another privatization approach is based on the Private Aggregation of Teacher Ensembles (PATE) framework (Papernot et al., 2016; 2018), which trains multiple teacher models on private data, and updates the generator with differentially private aggregation of teacher model outcomes (Jordon et al., 2018; Long et al., 2021; Wang et al., 2021a). GS-WGAN (Chen et al., 2020) leverages both DP-SGD and PATE, where multiple teacher models are trained as in PATE, while their outcomes are processed with gradient cliiping as in DP-SGD. We design PFGuard to complement these private GANs by also achieving fairness in data generation.

**Fair-only Data Generation**  The goal of model fairness is to avoid discriminating against certain demographics (Barocas et al., 2017; Feldman et al., 2015; Hardt et al., 2016), and fair data generation solves this problem by generating synthetic data to remove data bias. The main approaches of fair data generation are as follows: 1) modifying *training objectives* to balance model training (Xu et al., 2018; Sattigeri et al., 2019; Yu et al., 2020; Choi et al., 2020; Teo et al., 2023) and 2) modifying *latent distributions* of the input noise to obtain fairer outputs (Tan et al., 2020; Humayun et al., 2021). In comparison, PFGuard 1) modifies *sampling procedures* to balance model training while preserving original training objectives and 2) makes the key contribution of satisfying both privacy and fairness. There is another recent line of work using generated data together with original training data for model fairness (Roh et al., 2023; Zietlow et al., 2022), but they focus on classification tasks and assume to use given generative models.

**Privacy-Fairness Intersection**  Recent studies have shown that achieving DP can hurt model fairness in classification tasks (Bagdasaryan et al., 2019; Farrand et al., 2020; Xu et al., 2020; Esipova et al., 2022), decision-making processes (Pujol et al., 2020), and generation tasks (Cheng et al., 2021; Ganev et al., 2022; Bullwinkel et al., 2022; Rosenblatt et al., 2024). In addition, many studies have investigated the privacy-fairness-utility tradeoff, showing that achieving both privacy and fairness will necessarily sacrifice utility (Cummings et al., 2019; Agarwal, 2021; Sanyal et al., 2022). In comparison, our study uncovers the *counteractive* nature of privacy and fairness – achieving DP can compromise model fairness and achieving model fairness can compromise DP.

**Private and Fair Classification**  To effectively achieve both privacy and fairness in model training, various techniques have been developed for classification tasks (Jagielski et al., 2019; Xu et al., 2019; 2020; Tran et al., 2022; Esipova et al., 2022; Kulynych et al., 2022; Yaghini et al., 2023; Lowy et al., 2023). In comparison, PFGuard focuses on data generation tasks, which aim to learn the underlying training data distributions to generate synthetic data, and specifically tailors its fair training phase

to generative modeling objectives. Although having the fundamental different problem settings, we make comparisons with existing classification techniques that also leverage 1) importance sampling or 2) PTEL, which are the two key components of PFGuard. (Kulynych et al., 2022) also extends importance sampling to private settings and evaluate fairness in downstream classification. In contrast, PFGuard evaluates fairness in both synthetic data generation and downstream classification. (Yaghini et al., 2023) and (Tran et al., 2022) use PTEL mechanisms, but rely on public datasets to train student classifiers. In contrast, PFGuard eliminates the need for public datasets by making PTEL queries with generated samples from the student generator. We finally note that (Lowy et al., 2023) introduces the first DP fair learning method with convergence guarantees for empirical risk minimization. In contrast, PFGuard provides convergence guarantees for fair generative modeling.

