# OpenReview forum: "PFGuard: A Generative Framework with Privacy and Fairness Safeguards"
_ICLR.cc/2025/Conference — ICLR 2025 Poster_

### Official Review · Reviewer_vffx · 2024-11-01

**Soundness:** 4
**Presentation:** 3
**Contribution:** 4
**Rating:** 8
**Confidence:** 3

**Summary:**

In this work, the authors study the intersections of fairness and privacy in generative models, and show that these two goals can come into conflict. Furthermore, they highlight that naively combining existing fairness and privacy methods can fail. They propose PFGuard, a framework for achieving fairness and privacy simultaneously and then conduct experiments validating their method.

Overall, I think this is a paper with great contributions and analysis. I think some of the boldest claims should be softened, but otherwise I think it is a good candidate for ICLR.

**Strengths:**

Strengths:
 * I really liked the discussion of how fairness and privacy can come into conflict. I think it is somewhat "folklore" in the trustworthy ML community so I appreciated a more detailed discussion of the topic, especially in Section 3.
 * The PFGuard framework is a great solution to the problem. "Isolating" the fair training component and then distilling this knowledge privately is a great approach that I could see being taken to do many different downstream applications that also require privacy!
 * I think it is good the authors acknowledge that fairness adjusts sensitivity, which can still be made private with more noise, but point out the amount of noise to add may be unclear or performance degrading.

**Weaknesses:**

Weaknesses:
 * The authors focus on GANs as opposed to Diffusion Models, lessening the contemporary impact of the paper. This isn't necessarily a weakness, more an observation.
 * In Section 3, you start with **Adding Fairness Can Worsen Privacy** and describe in text some vague settings where fairness and privacy can come into conflict. I really wanted to see an elaboration or concrete example of how this could happen. I was very excited to see this in **Adding Privacy Can Worsen the Fairness-Utility Tradeoff** part! I think this is a great contribution. However, I think in the second part, you actually discuss both how fairness can worsen privacy (how fairness can modify the sensitivity, C) and then how adding privacy can worsen fairness (clipping worsens fairness adjustments). You should split these up according to the headers you wrote by moving the first example up to the first section. I would rewrite these two sections and cut some of the vaguer writing in the first section, and spend some more time elaborating on these two examples as they are a great contribution that I would be interested in more discussion of.
 * You say in Remark 1 that your study is the first to reveal that fairness and privacy techniques can counteract each other. I think this is too bold of a claim, given works you cite such as [1] and [2] that explore the topic.
 * I wouldn't say giving each teacher *probabilistically* one sample of the minority group is enough data to expect the teacher to be adequately fair. How did you come to this heuristic, and how can you justify it?
 * There are no fairness guarantees offered by this method because of data resampling and teacher distillation. I think the link between having balanced minibatches and the fairness of the teachers is more clear, but I wonder how distillation affects bias or if there is any literature you could cite here.

Notation:
 * Definition 2.1 - I would explain that domain $\mathcal{D}$ is a dataset, thus why D, D' can differ by a single sample.
 * You should also describe what function we measure sensitivity over. Is it the loss? is it the gradient? is it the model outputs? This will make it much clearer how exactly fairness interventions impact sensitivity in Section 3.

[1]: Eugene Bagdasaryan, Omid Poursaeed, and Vitaly Shmatikov. Differential privacy has disparate
impact on model accuracy. Advances in neural information processing systems, 32, 2019.
[2]: Tom Farrand, Fatemehsadat Mireshghallah, Sahib Singh, and Andrew Trask. Neither private nor fair:
Impact of data imbalance on utility and fairness in differential privacy. In Proceedings of the 2020
workshop on privacy-preserving machine learning in practice, pp. 15–19, 2020.

**Questions:**

Questions:
 * "Additional fair sampling does not require additional training complexity compared to say adding a loss term for fairness": I would argue that adding a loss term that can be backpropagated over is much simpler than having to do your complex sampling procedure to construct balanced minibatches. Can you justify or elaborate on why fair sampling is less complex? I don't think this claim is core to your paper so I think you could also do without it. You mention it later in the paper as well.
 * In your extensions to unavailable sensitive attributes, why do you need to train your classifier on less data? Given one of the benefits of PFGuard is any methods you apply before the teacher-distillation step need not be private, I would expect the best thing to do from a fairness perspective is train the best, beefiest sensitive-attribute classifier possible for fair training.
 * In your figures, why do you apply [method + PFGuard]? Isn't PFGuard an end-to-end fairness + privacy method? From what I can tell in your results you are applying PFGuard to existing DP generative models. This makes it difficult for me to interpret your results, and the unique impact of PFGuard on your results.
 * My intuition tells me that privacy always comes at a cost to utility, and this teacher + distillation procedure should be even more noisy than traditional DP-SGD methods. Can you comment on why this is not the case in your results? Why isn't a more advanced privacy method that permits fairness also resulting in further costs to utility than a traditional fairness method? How much of a fairness drop (from teacher fairness to generator fairness) do we incur because of the private distillation?

---

> ### Author Response · Authors · 2024-11-21
>
> To Reviewer vffx (Response 1/3),
>
> Thank you for your thoughtful review and constructive feedback. We respond to each of your points below.
>
> ---
> > **[W1]** The authors focus on GANs as opposed to Diffusion Models, lessening the contemporary impact of the paper. This isn't necessarily a weakness, more an observation.
> ---
>
> Thank you for your important observation. We believe our paper **contributes to aligning responsible generative models with contemporary generative models that support high utility.** Compared to prior attempts on fair-and-private generative models that primarily address low-dimensional structured data, PFGuard contributes to scale on high-dimensional data such as images by effectively decoupling fair and private learning phases. While further progress is still required to achieve image quality comparable to current diffusion models, we hope our work can serve as a step toward bridging this gap.
>
> &nbsp;
>
> ---
> > **[W2]** In Section 3, you start … I was very excited to see this in Adding Privacy Can Worsen the Fairness-Utility Tradeoff part! I think this is a great contribution. However, I think in the second part, … You should split these up according to the headers you wrote by moving the first example up to the first section.
> ---
> We are very glad to see your comment and really appreciate it. As per your great suggestion, we **reorganized Section 3 and refined the writing.** We again thank you for your detailed feedback, which helped us to improve our manuscript.
>
> &nbsp;
>
> ---
> > **[W3]** You say in Remark 1 that your study is the first to reveal that fairness and privacy techniques can counteract each other. I think this is too bold of a claim, given works you cite such as [1] and [2] that explore the topic.
> ---
>
> We appreciate your viewpoint. Our intent was to say that we show counteractions in both directions: (1) privacy techniques undermining fairness and (2) fairness techniques compromising privacy (e.g., Figure 1). For example, [Bagdasaryan et al., 2019] and [Farrand, 2020] primarily focus on (1) but not (2). However, we agree that Remark 1 can be too bold; **we thus instead strengthened the discussion in the related work (Sec. F, highlighted in blue), removing Remark 1**.
>
> &nbsp;
>
> ---
> > **[W4]** I wouldn't say giving each teacher probabilistically one sample of the minority group is enough data to expect the teacher to be adequately fair. How did you come to this heuristic, and how can you justify it?
> ---
>
> We would like to first clarify that we suggest an **“upper bound” on the number of teachers (n_T) to expect at least some improvements in fairness (Section 4.2), rather than a heuristic for sufficient fairness performance**. We now respond to your questions as follows:
>
> - We justify this upper bound in Figure 4. Increasing the number of teachers beyond this upper bound leads to a noticeable decline in fairness performances, although it still outperforms the private-only baseline.
>
> - The suggested upper bound $\lfloor |\mathcal{D}|\min_{s \in \mathcal{S}} p_{\text{bias}} (s) \rfloor$ corresponds to the size of the smallest minority data group in the dataset. Since each teacher receives a randomly divided disjoint data partition, this upper bound ensures that each teacher probabilistically receives at least one data sample from the smallest minority data group.
>
> Based on your valuable feedback, we further clarified this point in our revision (Section 4.2, highlighted in blue).
>
> &nbsp;
>
> ---
> > **[W5]** There are no fairness guarantees offered by this method because of data resampling and teacher distillation. I think the link between having balanced minibatches and the fairness of the teachers is more clear, but I wonder how distillation affects bias or if there is any literature you could cite here.
> ---
>
> We appreciate your precise question. As you noted, our convergence guarantee is on “fair generative modeling of a balanced distribution” from biased training data, not on the fairness of the final generator due to the noisy DP distillation step. **While most of our expressions already clarify this point, we further refined two terms** based on your valuable feedback (Section 1, highlighted in blue).
>
> We also note that providing formal guarantees on knowledge distillation remains a challenge in generative settings, while notable attempts have been made in classification tasks  [Bassily et al., 2018]. However, these analyses in classification tasks are not easily extensible to generative tasks, which often involve an intricate interplay between generator and teacher models (e.g., adversarial training) that introduces additional complexities.
>
> Bassily et al., "Model-agnostic private learning.", NeurIPS 2018

---

> ### Author Response · Authors · 2024-11-21
>
> To Reviewer vffx (Response 2/3),
>
> ---
> > **[Notations]** Definition 2.1 - I would explain that domain D is a dataset, thus why D, D' can differ by a single sample.
> You should also describe what function we measure sensitivity over. Is it the loss? is it the gradient? is it the model outputs? This will make it much clearer how exactly fairness interventions impact sensitivity in Section 3.
> ---
> We reflected your points in our revision (Section 2 and Section C.1, highlighted in blue), refining notations and adding explanations on sensitivity. Sensitivity is measured over the training algorithm of the generator, as this is the target model we aim to ensure DP [Long et al., 2021]. We really appreciate your detailed comments in helping us improve our manuscript.
>
> Long et al., "G-pate: Scalable differentially private data generator via private aggregation of teacher discriminators." NeurIPS 2021.
>
> &nbsp;
>
> ---
> > **[Q1]** "Additional fair sampling does not require additional training complexity compared to say adding a loss term for fairness": I would argue that adding a loss term that can be backpropagated over is much simpler than having to do your complex sampling procedure to construct balanced minibatches. Can you justify or elaborate on why fair sampling is less complex? I don't think this claim is core to your paper so I think you could also do without it.
> ---
> We respect your viewpoint and would like to clarify **our claim on complexity refers to “optimization complexity”**. While adding a fairness loss term can be simpler to implement as you noted, it can interfere with the main loss function, destabilizing training and requiring more training iterations to converge. In comparison, our sampling approach avoids such interference by preserving the main loss function.
>
> We also note that **PFGuard’s benefits in the optimization complexity lead to a more stable privacy guarantee in practice**. As shown in Table 3 and discussed in Section 5.2, baselines that alter loss functions lead to slower training convergence than PFGuard, introducing additional privacy cost (i.e., the more a model uses the data, the weaker privacy it provides).
>
> Based on your valuable feedback, we further clarified our expression on training complexity (Section 1, highlighted in blue).
>
> &nbsp;
>
> ---
> > **[Q2]** In your extensions to unavailable sensitive attributes, why do you need to train your classifier on less data? Given one of the benefits of PFGuard is any methods you apply before the teacher-distillation step need not be private, I would expect the best thing to do from a fairness perspective is train the best, beefiest sensitive-attribute classifier possible for fair training.
> ---
>
> We really appreciate your great question. You are right that using the best sensitive-attribute classifier can further improve fairness. However, **doing so essentially reduces the problem to the setting where sensitive labels are readily available.** Our goal was to extend PFGuard to constrained scenarios, so we designed the classifier to rely on less public reference data, ensuring applicability in more restrictive settings.
>
> &nbsp;
>
> ---
> > **[Q3]** In your figures, why do you apply [method + PFGuard]?
> ---
> Related to the previous response, another advantage of PFGuard is **flexibility with various advanced PTEL methods.** “[Method + PFGuard]” in Table 1 demonstrates the performance when integrated with different PTEL methods, where we use gradient sanitization [Chen et al., 2020] as the default method. The results show that PFGuard consistently enhances fairness while maintaining utility.
>
> Chen et al., "Gs-wgan: A gradient-sanitized approach for learning differentially private generators." NeurIPS 2020.

---

> ### Author Response · Authors · 2024-11-21
>
> To Reviewer vffx (Response 3/3),
>
> ---
> > **[Q4]** My intuition tells me that privacy always comes at a cost to utility, and this teacher + distillation procedure should be even more noisy than traditional DP-SGD methods. Can you comment on why this is not the case in your results? Why isn't a more advanced privacy method that permits fairness also resulting in further costs to utility than a traditional fairness method? How much of a fairness drop (from teacher fairness to generator fairness) do we incur because of the private distillation?
> ---
>
> Thank you for your insightful question. Our first observation is that **“how fairness intervention occurs” can be more critical than the choice of base privacy method (e.g., DP-SGD vs. teacher distillation).** For instance, two fairness-privacy methods using the same DP-SGD method can behave differently: DP-SGD-F may compromise both fairness and privacy, while DP-SGD-GA may trade fairness for utility (Table 3).
>
> Thus, to address your question, it is more reasonable to mainly compare the fairness intervention strategy, where **the key difference between prior methods and PFGuard is the "decoupling strategy."** Compared to prior methods that integrate fairness and privacy objectives within the same training phase, PFGuard’s decoupling strategy ensures that the private learning phase remains independent of fairness interventions. This decoupling, as previously discussed, provides more stable privacy guarantees, which we believe will be particularly beneficial for future generative models with complex training dynamics.
>
> &nbsp;
>
> We really appreciate your constructive and detailed feedback. Please let us know if your concern is not fully addressed. We are always happy to be engaged with you for further discussions.

---

> ### Comment · Reviewer_vffx · 2024-11-23
> **Response to Rebuttal**
>
> Thank you for the comprehensive response! You have answered many of my questions, I just wanted to follow up on a few things here:
>
>  * Upper bound on number of teachers: Thank you for this clarification. I think you should make it clear that this is the maximum number of teachers that would be feasible to consider at all, rather than a "recommended upper bound." It seems as though in practice to actually get fairness you will need to use many fewer teachers than this upper bound.
>  * Sensitivity definition: Thank you for clarifying sensitivity, but I still think you could be more clear -- specifically, is $f$ the function that achieves DP, or is $f$ some base function we wish to privatize (such as the loss or a gradient update) with a noise mechanism?
>  * "Decoupling strategy" in response to Q4. Thank you for this point! I think what you said earlier in response to W5 applies here. It would be important to acknowledge the fairness costs due to the formal privacy guarantees given by decoupling, namely that distillation might destroy fair behavior. However, looking at Table 3, it seems as though Fairness (reweighing) + Privacy (distillation) is performing better than just Fairness (reweighing). Can you explain this behavior? This seems very unexpected to me, as it says that conducting privacy actually improves the fairness and utility of the model.
>
> Thank you!

---

> ### Author Response · Authors · 2024-11-25
>
> To Reviewer vffx,
>
> We really appreciate your additional comments and great questions! We are happy to address them and respond to each point below.
>
> ---
> > **[P1]** Upper bound on the number of teachers
> ---
> You are right. Our suggestion refers to the “maximum number” of teachers, where the actual number of teachers can often be much lower in practice to ensure fairness. We opted to use mild expressions like “recommended” to account for potential randomness during training. As per your suggestion, **we revised the current expression to better reflect the meaning of the “maximum number” of teachers** (Section 4.2, highlighted in blue).
>
> &nbsp;
>
> ---
> > **[P2]** Sensitivity
> ---
>
> We really appreciate your question. Our sensitivity analysis **mainly refers to “f” as the “gradient”**, which ultimately leads to the sensitivity analysis of the generator training algorithm for a DP generator. Since the gradient $g$ serves as the base function in the generator training algorithm $\mathcal{A}$ (e.g., $\theta_G \leftarrow \theta_G - \eta \cdot g$ , where $\theta_G$ denotes the parameter of the generator $G$ and $ \eta$ denotes the learning rate), the sensitivity of the gradient function $g$ sufficiently captures the sensitivity of the training algorithm $\mathcal{A}$, which is the original function we want to ensure DP.
>
> However, Section 2 introduces the general definition of sensitivity, which may not clearly connect to our focus on gradients and the training algorithm. While we believe this general introduction of sensitivity is also important, we agree that the connection to our focus could be clearer. To effectively address your feedback, **we added a detailed explanation of how general sensitivity concepts apply to gradients in DP generator training** (Section C.1, highlighted in blue). We again appreciate your great question.
>
> &nbsp;
>
> ---
> > **[P3]** “RW+ DP noise (fairness+privacy)” performing better than “RW (fair-only)”
> ---
> We thank you for your interesting observation! Let us first clarify our results in Table 3, which is shown below. Here, “RW+DP noise” **achieves better fairness, but worse utility** compared to “RW (fair-only)” since both fairness and utility metrics are the lower the better.
>
> | Method           | Privacy ($\varepsilon$ ) | Fairness (KL ↓) | Fairness (Dist. Disp. ↓) | Utility (FID ↓) |
> |------------------|-------------|--------------|----------------|-----------|
> | RW (fair-only)	       | ✗           | 0.021         | 0.117           | 38.62      |
> | RW+ DP noise (fairness+privacy) | 13           |  0.009        |  0.044         | 106.94     |
>
> The improved fairness results from a **significant loss in utility, leading to uniformly low image quality across groups**. This result underscores how DP distillation can greatly alter the fairness-utility tradeoff of non-private training — either achieving fairness at the cost of utility or achieving utility at the cost of fairness — posing challenges for formal guarantees, as discussed in W5.
>
> &nbsp;
>
> We again appreciate all your valuable comments and thoughtful suggestions, which helped us to improve our manuscript. Please let us know if you have any additional concerns.

---

> > ### Comment · Reviewer_vffx · 2024-11-25
> >
> > Thank you for your responses! You have answered all of my questions and made my requested changes so I will be increasing my score. However, I am aware of the discrepancy between my score and the other reviewers and I am willing to discuss this with the AC/other reviewers.

---

> ### Author Response · Authors · 2024-11-26
>
> We would also like to express our special thanks to you. We learned a lot from your detailed feedback, and it was truly a pleasure to have constructive discussions with you.
>
> Warm regards,
>
> Authors

---

### Official Review · Reviewer_oBbq · 2024-11-01

**Soundness:** 2
**Presentation:** 3
**Contribution:** 1
**Rating:** 3
**Confidence:** 3

**Summary:**

This paper introduces PFGuard, a framework for privacy-preserving generative models that integrates fairness through a simple modification in the minibatch sampling process, specifically employing importance sampling based on group membership. The framework builds on PATE/PTEL, which is commonly used for privacy deep private generative models. The paper summarizes challenges in balancing privacy, utility, and fairness in generative models, then proposes the PFGuard framework (which relies on protected group-wise importance sampling), before finally empirically arguing that PFGuard achieves an improved balance of privacy and fairness.

**Strengths:**

This work quite successfully communicates the challenges in achieving the privacy/fairness tradeoff. I particularly appreciate the clarity of Figures 1, 2 and 3, along with the care with which the experimental results in Tables 3 and 4 are presented. Additionally, I believe that this is an interesting problem that deserves attention. I have closely read the main paper body, and appreciate the lack of obvious grammatical errors - overall the paper is well written.

**Weaknesses:**

Unfortunately, though the paper writing and presentation is of high quality and clarity, there are issues in terms of originality, significance and correctness.

W1: I’d like to highlight that Remark 1 (on novelty) is not correct and should be removed or further qualified. There appears to be ample prior work that “reveals how fairness and privacy techniques can counteract each other,” some in a more formal ways than this work (Bullwinkel et. al, https://arxiv.org/pdf/2205.04321 , Rosenblatt et. al https://arxiv.org/pdf/2312.11712 , Abroshan et. al (https://proceedings.mlr.press/v238/abroshan24a/abroshan24a.pdf , Cheng et. al https://dl.acm.org/doi/pdf/10.1145/3442188.3445879 )

W2: Additionally, Remark 2 either needs to be removed or needs further clarification - why would we extend classification techniques to the generative setting? Differentially private data generation (both synthetic tabular, see Mckenna et al 2019, 2024, https://proceedings.mlr.press/v97/mckenna19a/mckenna19a.pdf , https://arxiv.org/pdf/2201.12677 , Liu et. al 2021, 2023 https://proceedings.neurips.cc/paper_files/paper/2021/file/0678c572b0d5597d2d4a6b5bd135754c-Paper.pdf , https://proceedings.mlr.press/v202/liu23ag/liu23ag.pdf AND image Ghalebikesabi  et. al https://arxiv.org/pdf/2302.13861 ) is a very mature field, under much lighter assumptions then are necessary for PATE/PTEL.

W3: (W2) leads me to my main issue with the empirical results, which is that this is a bit of an apples to oranges comparison (or at least, it's not obvious that the comparison issue I see has been properly addressed). As you acknowledge, and as is the standard assumption with the PATE framework, we assume access to “ a public reference data on the order of 10%–100% of |D| for the estimation” (line 281). However, some comparisons in your paper (for example, in table 3) compare PFGuard directly to a method like DP-SGD (with further modifications), and for which it is not clear if the public reference data assumption is leveraged by the DP-SGD fit model (there are existing methods to help do this).

In fact, assuming an unbiased public reference sample is quite a strong assumption, and having no experiments on how a biased public reference sample would effect your results is questionable. Additionally, if I had access to a public reference sample, even only 10% of some large data sample |D|, why wouldn’t I just train on this sample? I’m missing where you note something about label missingness in this public sample, but maybe that isn’t the assumption? Is the assumption that we have full access to this public unbiased sample, and that its size is substantial? We would certainly want to compare to just training on that if this were the case.

W4: My 1 sentence summary of the proposed methodology here is as follows: take the PATE/PTEL framework, and during the sampling for teacher creation, use importance sampling based on group membership. This, in of itself, is a reasonable approach, but it is not clear to me how this work improves on the component parts of the PFGuard method (which are well established and well studied methods), besides using them in tandem. Nor does that work consider the model of importance sampling and how that might affect utility in other ways from a formal perspective, despite considerable prior work on the utility of PATE (Bassily et. al, https://arxiv.org/pdf/1803.05101 ). Nor does it contend formally with any potential privacy concerns (even to just dispel this concern with a short proof or proof adjustment). Given that I have some issues with the assumptions of the empirical results, this leads me to my score.

**Questions:**

All that said, I am open to raising my score slightly if the authors can adequately address the following questions,

Q1: How precisely is the public reference sample handled experimentally? Are all methods given “equal access” so to speak? If the sample is assumed to be complete (i.e. containing all relevant columns from D) also please provide a test on the same metrics you present for the private/fair methods of simply using the holdout sample at different subsample percentages on each task.

Q2: Why should we use importance sampling, instead of just constructing (potentially deterministically) balanced samples using stratified sampling? It seems believable (although is not proved here) that in the PATE/PTEL framework we can use whichever sampling technique we want, so long as it is randomized (although this is not explicitly proven by the authors or cited). However, given that, AND the assumption of access to the membership of samples in protected classes, importance sampling vs. stratified sampling should be explored. Or maybe I’m missing something.

Q3: Please, if you can, offer a more formal characterization of how importance sampling does not effect the privacy guarantee of PATE/PTEL. You can also use the framework presented in Bassily et. al .

---

> ### Author Response · Authors · 2024-11-21
>
> To Reviewer oBbq (Response 1/2),
>
> Thank you for your thoughtful review and constructive feedback. We respond to each of your points below.
>
> ---
> > **[W1]** I’d like to highlight that Remark 1 (on novelty) is not correct and should be removed or further qualified. There appears to be ample prior work that “reveals how fairness and privacy techniques can counteract each other,” some in a more formal ways than this work. [Bullwinkel et. al, 2022] ...
> ---
> We appreciate your viewpoint. Our intent was to say that we show counteractions in both directions: (1) privacy techniques undermining fairness and (2) fairness techniques compromising privacy. For example, the cited works primarily focus on 1) [Bullwinkel et al., 2022; Rosenblatt et al., 2024; Cheng et al., 2021], or addressing fairness alone [Abroshan et al., 2024]. However, we agree that Remark 1 can be too bold; **we thus instead strengthened the discussion in the related work (Sec. F, highlighted in blue), removing Remark 1.** We again thank you for your feedback, which helped us improve the manuscript.
>
> &nbsp;
>
> ---
> > **[W2]** Additionally, Remark 2 either needs to be removed or needs further clarification - why would we extend classification techniques to the generative setting?
> ---
> We appreciate your interesting point. Our intent of **Remark 2 is to consider DP-SGD, a widely used classification technique that is also widely used in generative models.** With recent fair variants of DP-SGD (e.g., DP-SGD-F [Xu et al., 2020]), applying these techniques to train generative models can provide a valid baseline for fair and private generative models. Additionally, such baselines may feel more natural to readers familiar with classification settings, like Reviewer pijU in this rebuttal.
>
> Xu et al., "Removing disparate impact on model accuracy in differentially private stochastic gradient descent.", ACM SIGKDD 2021.
>
> &nbsp;
>
> ---
> > **[W3]** As you acknowledge, and as is the standard assumption with the PATE framework, we assume access to “ a public reference data on the order of 10%–100% of |D| for the estimation” (line 281). However, some comparisons in your paper (for example, in table 3) compare PFGuard directly to a method like DP-SGD (with further modifications), and for which it is not clear if the public reference data assumption is leveraged by the DP-SGD fit model (there are existing methods to help do this). … Are all methods given “equal access” so to speak?
> ---
>
> We really appreciate your detailed feedback and thoughtful comments here, but we believe there may be a misunderstanding regarding **Line 281, which explains the fairness technique [Choi et al, 2020], not PATE [Papernot et al., 2017].** We believe this particular line may have influenced the subsequent points raised.
>
> We thus would like to clarify that **PTEL-based generative models we used in our experiments do not require any public data**, unlike PATE used in classification settings. We thus believe comparisons between PFGuard, PTEL-based generative models, and DP-SGD remain valid.
>
> Choi et al., "Fair generative modeling via weak supervision.", ICML 2020. \
> Papernot et al., "Semi-supervised knowledge transfer for deep learning from private training data.", ICLR 2017.
>
> &nbsp;
>
> ---
> > **[W3 & Q1]** In fact, assuming an unbiased public reference sample is quite a strong assumption, and having no experiments on how a biased public reference sample would effect your results is questionable.
> ---
> We again thank you for your valuable comment. To compare with the aforementioned fairness technique [Choi et al, 2020], which supports extensions to unknown sensitive settings by using public data, we also analyzed PFGuard’s performance under the same conditions. While we explicitly distinguished such cases that allow public data and specified the size used (denoted with “perc” in Table 3), we agree that the current baseline ordering shown in Table 3 can be misleading. **We thus revised Table 3 to more clearly separate cases that allow public data.** Additionally, since [Choi et al, 2020] only requires “balanced” public data to serve as the reference data, we do not analyze performance with “biased” public data.
>
> We hope this clarification addresses your concerns on our experimental results. Please let us know if your concern is not fully addressed.
>
> Choi et al., "Fair generative modeling via weak supervision.", ICML 2020.
>
> &nbsp;
>
> ---
> > **[W3 & Q1]** Additionally, if I had access to a public reference sample, even only 10% of some large data sample |D|, why wouldn’t I just train on this sample? … We would certainly want to compare to just training on that if this were the case. Please provide a test …
> ---
>
> Your suggestion is valid, **but note that it reduces to fairness-only data generation**; only the public dataset is used, and no private sensitive data is involved. We also note that this scenario is already included in our results as a baseline (Table 3, denoted as “Fair-only”).

---

> ### Author Response · Authors · 2024-11-21
>
> To Reviewer oBbq (Response 2/2),
>
> ---
> > **[W4]** Nor does that work consider the model of importance sampling and how that might affect utility in other ways from a formal perspective, despite considerable prior work on the utility of PATE (Bassily et. al, 2018).
> ---
>
> Continuing from the above comment, we would like to clarify that **the scope of [Bassily et al., 2018] is different from our paper’s scope**; [Bassily et al., 2018] focuses on classification tasks to achieve utility and privacy, while we focus on generative tasks to achieve utility, privacy, and fairness. Thus, while we agree that formal utility guarantees like those in [Bassily et al., 2018] are always desirable, they can be **highly challenging in our setup** given the (1) complexity of model architectures that include adversarial training such as GANs, and (2) the interplay of multiple objectives (i.e., fairness, utility, privacy) that can affect each other. We thus put more effort into empirical validation to analyze how importance sampling might affect utility, including image quality, downstream classification accuracy, and computational time.
>
> Bassily et al., "Model-agnostic private learning.", NeurIPS 2018
>
> &nbsp;
>
> ---
> > **[Q2]** Why should we use importance sampling, instead of just constructing (potentially deterministically) balanced samples using stratified sampling? ... However, given that, AND the assumption of access to the membership of samples in protected classes, importance sampling vs. stratified sampling should be explored.
> ---
>
> Thank you for the interesting question. We would like to clarify that **stratified sampling and importance sampling have fundamentally different statistical goals**. Stratified sampling preserves the original distribution, while importance sampling adjusts the sample to match a different target distribution. For fairness, importance sampling can be better suited, as it balances the data distribution instead of reproducing the original biased distribution. Additionally, if sensitive attributes are used in stratified sampling to create balanced data as you noted, **it no longer aligns with the purpose of stratified sampling and essentially reduces to importance sampling.**
>
> &nbsp;
>
> ---
> > **[Q3]** Please, if you can, offer a more formal characterization of how importance sampling does not effect the privacy guarantee of PATE/PTEL.
> ---
>
> We do appreciate your comment and **newly added theoretical proofs (Sec. C.1., highlighted in blue)** showing how PFGuard preserves the sensitivity of PTEL methods, complementing natural language explanations provided in the previous manuscript. We again thank you for your constructive feedback.
>
> &nbsp;
>
> Please let us know if your concern is not fully addressed. We are always happy to be engaged with you for further discussions.

---

> ### Author Response · Authors · 2024-11-25
> **Looking forward to hearing from you**
>
> We understand that this is a busy time for everyone. We would be grateful to know whether our response has addressed your concerns. Please feel free to let us know if you have any remaining questions.
>
> Thank you,
>
> Authors

---

> > ### Comment · Reviewer_oBbq · 2024-11-25
> >
> > I appreciate the authors efforts during the rebuttal process. However, after reviewing the other reviewers comments, and considering the authors rebuttal, I have decided to maintain my score. I encourage the authors to consider a more formal justification for why I would prefer this importance sampling method compared to simply choosing the disjoint subsets based on group membership, and more empirical results justifying that decision as well.

---

> ### Author Response · Authors · 2024-11-27
>
> We really appreciate your comment on the remaining concerns, where we may have misunderstood your previous suggestion due to the terminology of stratified sampling. **Your suggestion – choosing disjoint subsets based on group membership – is highly valuable for comparison with Importance Sampling (IS)**, as this strategy may lead to better representation of data groups by training teachers exclusively on each data group.
>
>
> **However, this sampling strategy can have two theoretical downsides: (1) downgraded utility of teacher ensemble and (2) increased privacy cost.** Training teachers on non-i.i.d. datasets can lead to inconsistent convergence across teachers, leading to low consensus during teacher voting [Dodwadmath et al., 2022]. This low consensus can reduce the overall prediction accuracy (i.e., utility) of the teacher ensemble [Dodwadmath et al., 2022], and also increase privacy costs during DP aggregation of teacher votes [Papernot et al., 2018].
>
>
> **We empirically demonstrate choosing disjoint subsets w.r.t. groups can result in a suboptimal privacy-fairness-utility tradeoff.** As shown in the table below, this strategy achieves a similar level of fairness compared to IS, but significantly reduces utility, which aligns with the above theoretical observations.
>
> | Method           | Privacy ($\varepsilon$ ) | Fairness (KL ↓)) | Fairness (Dist. Disp ↓) | Utility (FID ↓) |
> |------------------|-------------|--------------|----------------|-----------|
> | Privacy-only	       |     10       | 0.177         | 0.383          | 77.97    |
> | Privacy + subset w.r.t. groups | 10          |  0.090           | 0.209      | 135.35     |
> | Privacy + IS (ours)  | 10          |  0.067   |  0.242        | 83.67    |
>
> **In our revision, we added the above discussion to support the benefits of IS in terms of i.i.d. training distribution (Section E.5, highlighted in blue).** We really appreciate your valuable suggestion, which helped us to improve our manuscript. Please let us know if there are remaining concerns, and we are happy to be engaged with you for further discussions.
>
>
> Dodwadmath et al.,  "Preserving privacy with PATE for heterogeneous data.", NeurIPS Workshop on Distribution Shifts 2022 \
> Papernot et al., "Scalable private learning with pate.", ICLR 2018

---

### Official Review · Reviewer_WKsH · 2024-11-03

**Soundness:** 3
**Presentation:** 3
**Contribution:** 2
**Rating:** 6
**Confidence:** 4

**Summary:**

This paper focuses on data privacy and model fairness for developing generative models. They claim natively combining differential privacy and fairness learning techniques may cause conflicts, and then propose PFGuard framework to simultaneously balance the two objective and also the utility. The core insight is to employ an ensemble of multiple teacher models. And they also do experiments on GANs with images datasets as benchmarks to show a better trade-off can be achieved.

**Strengths:**

1. I agree with the authors that the conflicts between data privacy and fairness do exist for generative models. It is an interesting research problem to be explored by research community.
2. I think importance resampling can be a useful approach to get balanced training data, which is helpful to train a fair generative model.
3. The experiments shown in tables demonstrated that better utility and fairness are achieved by fixing a privacy budget.

**Weaknesses:**

1. The authors have claimed the interested privacy of this paper is to defend against training data reconstruction, while naive differential privacy techniques preserve the membership instead of data content. This has been recognised in the early work [1] but ignored by this paper.
2. Since only a smaller training data is derived after resampling for fairness, this may degrade the quality of generated data, as claimed in [2]. In this case, the quality of generated data will be sacrificed.
3. It is easy to understand that minority samples should be upweighted in fairness. But in what scenarios minority samples should be downweighted for DP? Because sensitivity controls the strength of the added noise, maybe how will minority samples affect the sensitivity should be explained.
4. In Fig. 3, reweighted and rewerighting are both used.  Are they same? In addition, it seems privacy budget is fixed and then the trade-off between utility and fairness is studied, and so as to experiments. This is not aligned with the motivation where fairness and privacy conflicts.
5. The proposed framework is a sequential procedure where the first component is about fairness while the second one is for privacy. In this sense, it is not very convincing to say they two can be better traded off.
6. I would like to see some Pareto front results in terms of three metrics, which can better demonstrate the proposed method.

[1] Bounding Training Data Reconstruction in Private (Deep) Learning, ICML 2022.
[2] Generative Adversarial Ranking Nets, JMLR 2024.

**Questions:**

Please refer to Weaknesses.

Also, can you justify the novelty of using PTEL in this work? Because this is the main technique of the proposed framework.

---

> ### Author Response · Authors · 2024-11-21
>
> To Reviewer WKsH (Response 1/2),
>
> Thank you for your thoughtful review and constructive feedback. We respond to each of your points below.
>
> ---
> > **[W1]** The authors have claimed the interested privacy of this paper is to defend against training data reconstruction, while naive differential privacy techniques preserve the membership instead of data content.
> ---
> We appreciate your valuable viewpoint. We would like to clarify that mentioning “training data reconstruction” is **not to limit our scope to reconstruction-specific defenses**, but to introduce privacy concerns in generative models. As you note, our focus is on DP techniques, which are conventionally used to address these privacy concerns and can also mitigate the mentioned reconstruction risks [Stock et al., 2022].
>
> Nevertheless, we agree with your point that DP techniques are more strongly associated with membership inference than data reconstruction. **We thus refined our expression to “leakage of personal sensitive information” (Section 1, highlighted in blue)**, reflecting the broader aim of DP in preventing information leakage.
>
> Stock et al., "Defending against reconstruction attacks with Renyi differential privacy.", arXiv 2022.
>
> &nbsp;
>
> ---
> > **[W2]** Since only a smaller training data is derived after resampling for fairness, this may degrade the quality of generated data, as claimed in [2]. In this case, the quality of generated data will be sacrificed.
> ---
>
> We would like to clarify that **PFGuard can improve the overall data quality by significantly enhancing image quality for the minority data group.** While your point is valid for the *majority data group* – PFGuard indeed samples fewer data samples from these groups compared to random sampling as you said – the quality gains for the minority data group can often outweigh the quality losses for the majority data group, resulting in improvements in overall quality (Table 1, Table 6). In addition, performance improvements in downstream tasks (Table 2, Table 4) further support that PFGuard can be beneficial for enhancing the overall utility of generated data.
>
> &nbsp;
>
> ---
> > **[W3]** It is easy to understand that minority samples should be upweighted in fairness. But in what scenarios minority samples should be downweighted for DP? Because sensitivity controls the strength of the added noise, maybe how will minority samples affect the sensitivity should be explained.
> ---
> Thank you for the insightful question. We would like to first clarify that **we do discuss how minority samples affect the sensitivity in Section 3.** We then explain scenarios that can require downweighting of minority samples.
>
> **In Section 3, Figure 3 illustrates how minority samples with large gradients can affect sensitivity, leading to high noise.** Since sensitivity measures the maximum impact of “any” data sample, large gradients of minority samples lead to high sensitivity values, leading to high DP noise as you noted. In order to prevent such high noise undermining the utility, downweighting minority samples with large gradients is often necessary to balance the privacy-utility tradeoff.
>
> **In practice, these disproportionately large gradients of minority data can happen due to several reasons:** 1) DP noise causing imbalanced convergence speeds, leading to slower convergence and higher gradients for minority groups [Bagdasaryan et al., 2019; Farrand et al., 2020] or 2) fairness adjustments that upweight minority gradients to balance learning between data groups. Our paper newly demonstrates 2) as a fairness-privacy conflict, where fairness upweights minority gradients, but privacy again downweights them to prevent excessive DP noise.
>
>
> Bagdasaryan et al., “Differential privacy has disparate impact on model accuracy”, NeurIPS 2019. \
> Farrand et al., “Neither private nor fair: Impact of data imbalance on utility and fairness in differential privacy.”, PPMLP’20

---

> ### Author Response · Authors · 2024-11-21
>
> To Reviewer WKsH (Response 2/2),
>
> ---
> > **[W4]** In Fig. 3, reweighted and rewerighting are both used. Are they same?
> ---
> **We revised Figure 3 to use “reweighting”, reflecting your point.** We appreciate your feedback in helping us improve our manuscript.
>
> &nbsp;
>
> ---
> > **[W4 & W6]**  It seems privacy budget is fixed and then the trade-off between utility and fairness is studied, and so as to experiments. This is not aligned with the motivation where fairness and privacy conflicts. I would like to see some Pareto front results in terms of three metrics, which can better demonstrate the proposed method.
> ---
> Your suggestion of Pareto frontier results is great; **we now added two new experimental results including Pareto Frontier results (Section E.1, highlighted in blue).** These results show more general observations on the privacy-fairness-utility tradeoff when epsilon varies, where PFGuard consistently outperforms baseline methods.
>
> In addition, we would like to explain that **the reason we fixed the privacy budget is to clearly show the fairness-privacy conflict under constrained conditions (e.g., a limited number of iterations).** For example, the fixed budget allows observations like (1) how much the relative privacy cost can be incurred to ensure fairness (Table 3 shows this cost can be at most 30%) or (2) how the behaviors of baseline methods can be different under limited privacy budgets (Table 3 shows DP-SGD-F can compromise fairness and privacy, where DP-SGD-GA can compromise fairness and utility).
>
> &nbsp;
>
> ---
> > **[W5]** The proposed framework is a sequential procedure where the first component is about fairness while the second one is for privacy. In this sense, it is not very convincing to say they two can be better traded off.
> ---
>
> We respect your viewpoint and would like to highlight below two points.
>
> **We believe that PFGuard has a clear distinction with a naive sequential design, which may fail to fully decouple fairness and privacy even though the training phases are separated.** As detailed in Section 3, such designs often result in entangled dynamics where fairness interventions affect privacy guarantees (e.g., through sensitivity variations or additional noise). In contrast, PFGuard ensures that the privacy analysis remains independent of fairness interventions, supporting new advantages like high modularity with various PTEL methods without additional privacy concerns.
>
> **We also demonstrate how effectively decoupled design can lead to a better fairness-privacy-utility tradeoff compared to integrated design.** While recent fairness-privacy approaches aim to achieve two objectives in the same training phase (e.g., DP-SGD with fairness constraints), we identify the inherent conflict between fairness and privacy (illustrated in Figure 1) that can undermine each objective. Our experiments (Table 3) show that these approaches can lead to additional privacy costs while having theoretically valid guarantees (e.g., DP-SGD-F), or high unfairness (e.g., DP-SGD-GA). In contrast, PFGuard's design demonstrates a more stable privacy guarantee and utility while enhancing fairness.
>
> &nbsp;
>
> ---
> > **[Q1]** Also, can you justify the novelty of using PTEL in this work? Because this is the main technique of the proposed framework.
> ---
>
> To effectively address your question, let us **make a comparison with DP-SGD [Abadi et al., 2016], which directly trains the target model on private sensitive data.** Here, fairness interventions can also directly affect the target model and its privacy guarantee. For example, applying PFGuard's fair sampling with DP-SGD could repeatedly feed certain data samples to the target model, weakening privacy guarantees.
>
> In contrast, **PTEL trains the target model using only teacher models, instead of directly using private sensitive data.** Our key idea is to leverage this point to decouple fairness and privacy. We apply fairness interventions at the teacher level – which will not directly affect the target model – and then privatize the knowledge transfer stage to provide a strict DP guarantee to the target model even with the fairness intervention. With this design, note that teachers do not necessarily require private training [Chen et al., 2023]; we can thus effectively train fair but non-private teachers, eliminating the need to achieve both objectives in the same training phase, which can conflict with each other as we identified in Figure 1.
>
> Abadi et al., "Deep learning with differential privacy.", ACM SIGSAC 2016.
> Chen et al., "A unified view of differentially private deep generative modeling.", arXiv 2023.
>
> &nbsp;
>
> We again thank you for your constructive feedback and please let us know if your concern is not fully addressed. We are always happy to be engaged with you for further discussions.

---

> ### Author Response · Authors · 2024-11-25
> **Looking forward to hearing from you**
>
> We understand that this is a busy time for everyone. We would be grateful to know whether our response has addressed your concerns. Please feel free to let us know if you have any remaining questions.
>
> Thank you,
>
> Authors

---

> > ### Comment · Reviewer_WKsH · 2024-11-26
> >
> > Thanks for your response. I think now I better understand your work.
> >
> > I personally do not buy the insight of "teachers do not necessarily require private training [Chen et al., 2023]", because GANs train both discriminator and generator simultaneously, unless you can justify in some scenarios that teacher ensemble training part can be unobservable for adversaries.
> >
> > By checking the proposed framework, I found a very similar work [1] from existing literature, which was not mentioned in the paper. I understand [1] did not focus on generation tasks, but now I doubt the faithfulness of this paper.
> >
> > I also have one more suggestion. Since the proposed method is under GAN, I think more explanations towards fairness and privacy can be added from the adversarial perspective.
> >
> > Despite such concerns, I would like to raise my rating to 5 based on the quality of the current version.
> >
> > [1] A Fairness Analysis on Private Aggregation of Teacher Ensembles. AAAI 2022.

---

> ### Author Response · Authors · 2024-11-27
>
> We truly appreciate you raising the score and sharing your additional comments with us. We are happy to address them and respond to each point below.
>
> ---
> > Justification of scenarios
> ---
> Thank you for your feedback. We would like to clarify that **we follow the conventional setup in DP generative models** [Chen et al., 2020; Long et al., 2021; Wang et al., 2021], where (1) only the generator is released publicly, and (2) teacher models are kept private thus inaccessible by adversaries. **While we do mention this point in Section 4.2, we revised our main figure of the framework and added more citations** to reflect your valuable feedback (Figure 2 and Section 4.2, highlighted in blue).
>
>
> Chen et al., "Gs-wgan: A gradient-sanitized approach for learning differentially private generators.", NeurIPS 2020. \
> Long et al., "G-pate: Scalable differentially private data generator via private aggregation of teacher discriminators.", NeurIPS 2021. \
> Wang et al., "Datalens: Scalable privacy preserving training via gradient compression and aggregation.", ACM SIGSAC 2021.
>
> &nbsp;
>
> ---
> > Similar work of [Tran et al., 2022]
> ---
> We do appreciate your detailed comment, but **the cited paper [Tran et al., 2022] appears to be the preprint version of [Tran et al., 2023], which we compare in detail in Section A.** As you noted, their work focuses on classification settings, and extending their work to generation settings is not straightforward due to the difference in DP notions; [Tran et al., 2023] focuses on a DP notion that protects sensitive attributes, whereas we address a general DP notion that protects all data attributes.
>
> Tran et al. "SF-PATE: scalable, fair, and private aggregation of teacher ensembles.", IJCAI 2023.
>
> &nbsp;
>
> ---
> > Adversarial training
> ---
> Thank you for your insightful suggestion! As you noted, **we believe that the goal of balancing fairness and privacy can naturally align with adversarial training components: (1) the min-max game and (2) the optimal discriminator.**
>
> Adversarial learning optimizes conflicting goals (i.e., the *min-max game*) using two components: the generator and the discriminator. Based on our intuition that privacy and fairness can also have conflicting goals, assigning privacy and fairness to each component and using adversarial learning may more effectively balance the fairness-privacy conflict. Moreover, since the convergence of adversarial training greatly depends on the *optimality of the discriminator*, assigning fairness specifically to the discriminator may effectively teach the generator an optimal state that is also fair.
>
> Based on your valuable suggestion, **we recognized that PFGuard design aligns with both (1) and (2)**, by assigning fairness to teacher models (i.e., discriminators) and privacy to the target generator. While PFGuard supports various loss functions and does not necessarily require adversarial training, we think PFGuard can be particularly synergistic with adversarial training. We really appreciate your insightful suggestion and added this dicussion in our revision (Section C.3, highlighted in blue).
>
> &nbsp;
>
> We again thank you for all your valuable comments, which helped us to improve our manuscript. Please let us know if you have any additional concerns.

---

> > ### Comment · Reviewer_WKsH · 2024-12-03
> >
> > Thank you for your response. While DP guarantees are not typically verified through experiments, my understanding is that the core privacy guarantee relies on the principle that even if an adversary has sufficient prior knowledge to replicate the entire training process, they still cannot confidently determine whether a specific sample was included in the training data.
> >
> > It seems, however, that the authors are operating under the assumption that the adversary only has access to the public generators in practice. While this may be acceptable, I believe it somewhat relaxes the strict DP concept.
> >
> > After reviewing the comments from other reviewers, I find that the current version of the paper is in good shape overall, and I have decided to raise my score to 6.

---

> ### Author Response · Authors · 2024-12-03
>
> We truly appreciate you raising the score. **We also agree that DP generative models are in the perspective of a “data owner”**, who is assumed to have the “choice” of what to release to the adversary [Chen et al., 2023]. As you correctly mentioned, these assumptions can be diverse; early works like DP-GAN [Xie et al., 2018] assumed that the data owner releases all training parameters -- including those of teacher models -- while the following works used a different assumption that the data owner releases only the “generator”, enabling a better privacy-utility tradeoff.
>
> We thus would like to specially thank you for your feedback on DP scenarios, **which can be diverse and can affect the maximum privacy-utility tradeoff**. We believe the used DP scenarios became much clearer in our revision by having discussion with you, and we will also add the above discussion of the "data owner" to further clarify our scope. Thank you for sharing your valuable feedback with us.
>
> Chen et al., "A unified view of differentially private deep generative modeling.", arXiv 2023. \
> Xie et al., "Differentially private generative adversarial network.", arXiv 2018.

---

### Official Review · Reviewer_sxpU · 2024-11-10

**Soundness:** 3
**Presentation:** 3
**Contribution:** 3
**Rating:** 8
**Confidence:** 4

**Summary:**

This paper studies tensions between fairness and privacy in generative models. It proposes PFGuards, a framework leveraging an ensemble of teacher models to jointly enforce fairness and differential privacy while preserving utility. The proposed method consists of training an ensemble of fair teachers using balanced minibatch sampling and using their differentially private (leveraging the PATE framework) aggregated output to train a privatized generator. Experiments demonstrate improved performances in fairness and utility compared to existing approaches.

**Strengths:**

- Good quality of the presentation
- Extensive experiments on MNIST, FashionMNIST, and CelebA
- Comparison against several baselines, including fair variants of DP-SGD

**Weaknesses:**

- Limited choice in epsilon values, making it difficult to understand the effect of privacy budget on utility and fairness
- The choice of datasets and bias settings makes it difficult to determine whether the method would perform effectively in real-world scenarios. I recommend incorporating tabular datasets, as done in (Tran et al., 2021b), to assess group fairness. Additionally, it would be valuable to explore alternative fairness notions for datasets where group fairness may not be applicable

**Questions:**

- Is there any consistent trend in utility and fairness as the privacy budget increases? I would suggest the authors to perform a similar analysis as in (Tran et al., 2021b -- Fig. 2)
- Given the high-dimensional context, other fairness notions, such as Rawlsian Max-Min fairness, could be more appropriate. Can the framework proposed extend to such notions?

---

> ### Author Response · Authors · 2024-11-21
>
> To Reviewer sxpU (Response 1/2),
>
> We appreciate your valuable review and constructive feedback. We respond to each of your points below.
> &nbsp;
>
> ---
> > **[W1 & Q1]** Limited choice in epsilon values, making it difficult to understand the effect of privacy budget on utility and fairness. Is there any consistent trend in utility and fairness as the privacy budget increases? I would suggest the authors to perform a similar analysis as in (Tran et al., 2021b -- Fig. 2)
> ---
> We really appreciate your great suggestion. **We now added two new experiments (Section E.1, highlighted in blue) to analyze the privacy-fairness-utility tradeoff when varying epsilon values.** In particular, we included analyses similar to those in [Tran et al., 2021b] as per your great suggestion.
>
> Interestingly, **our results reveal two distinct trends depending on the fairness criteria**. For one fairness criteria to balance data quantity w.r.t. groups, we observed a consistent trend, where stronger privacy constraints lead to both downgraded utility and worse fairness. In contrast, for another fairness criteria to balance data quality w.r.t. groups, we observe stronger privacy constraints can lead to downgraded utility with low image quality, but having better fairness with uniformly low image quality w.r.t. groups.
>
> We added a more detailed analysis along with the experiment results in our revision. We again appreciate your great question, which has improved our manuscript.
>
> &nbsp;
>
> ---
> > **[W2 & Q2]** The choice of datasets and bias settings makes it difficult to determine whether the method would perform effectively in real-world scenarios. I recommend incorporating tabular datasets, as done in (Tran et al., 2021b), to assess group fairness.
> ---
>
> We value your suggestion and **added a new experiment (Section E.2) to show how PFGuard can also support tabular data as well as image data.** We observe that PFGuard 1) greatly improves fairness compared to privacy-only baselines, with only a slight utility tradeoff, and 2) achieves better fairness while maintaining comparable utility compared to the fairness-privacy baseline (FFPDG).
>
> In addition, we would like to explain that tabular data is not our main focus. We believe that a key contribution of PFGuard is its scalability to high-dimensional data such as images, which has not yet been addressed by prior works with their primary focus on low-dimensional tabular data. We thus explore real-world scenarios with image data such as the CelebA dataset, following prior works [Long et al., 2021; Wang et al., 2021a].
>
> | Method           | Privacy ($\varepsilon$) | Fairness (EO Disp ↓)) | Fairness (Dem. Disp. ↓) | Utility (AUROC ↑) |
> |------------------|-------------|--------------|----------------|-----------|
> | Vanilla          | ✗           | 0.56         | 0.58           | **0.80**      |
> | Fair-only        | ✗           | **0.07**         | **0.07**           | 0.75      |
> | DP-only (DP-WGAN)| 1.0           | 0.31         | 0.30           | 0.69      |
> | DP-only (PATE-GAN)| 1.0          | 0.19         | 0.22           | 0.74      |
> | DP-only (RON-Gauss)| 1.0         | 0.18         | 0.14           | 0.70      |
> | FFPDG            | 1.0           | 0.12         | 0.20           | 0.75      |
> | **PFGuard**      | 1.0           | *0.08*     | *0.12*       | *0.76*  |
>
> Long et al., "G-pate: Scalable differentially private data generator via private aggregation of teacher discriminators.", NeurIPS 2021. \
> Wang et al., "Datalens: Scalable privacy preserving training via gradient compression and aggregation.", ACM SIGSAC 2021.

---

> ### Author Response · Authors · 2024-11-21
>
> To Reviewer sxpU (Response 2/2),
>
> ---
> > **[Q3]** It would be valuable to explore alternative fairness notions for datasets where group fairness may not be applicable. Given the high-dimensional context, other fairness notions, such as Rawlsian Max-Min fairness, could be more appropriate. Can the framework proposed extend to such notions?
> ---
> Thank you for your interesting question. Fortunately, **PFGuard supports integrating existing methods for Rawlsian Max-Min fairness while preserving privacy guarantee if they meet two conditions**: 1) applied to teacher models to avoid direct impact on the target generator, and 2) maintain data disjointness where one sample affects only one teacher, which is a foundation of PFGuard’s privacy guarantee (detailed in Sec. 4.2).
>
>
> For example, **GOLD [Mo et al., NeurIPS’19] is compatible with PFGuard** by satisfying both conditions. GOLD achieves Rawlsian Max-Min fairness by 1) using log density ratio estimates to identify worst-group samples and 2) reweighting the discriminator (teacher) loss to improve performance on these samples. Thus, GOLD applies to teacher models (condition 1) and maintains data disjointness (condition 2), as one reweighted sample affects only one discriminator.
>
> Inspired by your question, we added a paragraph to highlight PFGuard’s compatibility with other methods, specifying the above two conditions (Section C.3, highlighted in blue). We again thank you for your great question.
>
> Mo, Sangwoo, et al. "Mining gold samples for conditional gans." NeurIPS 2019.
>
> &nbsp;
>
> We again appreciate your feedback in helping us improve the manuscript, and please let us know if your concern is not fully addressed. We are always happy to be engaged with you for further discussions.

---

> ### Author Response · Authors · 2024-11-25
> **Looking forward to hearing from you**
>
> We understand that this is a busy time for everyone. We would be grateful to know whether our response has addressed your concerns. Please feel free to let us know if you have any remaining questions.
>
> Thank you,
>
> Authors

---

> > ### Comment · Reviewer_sxpU · 2024-11-25
> > **Feedback**
> >
> > Thank you for providing the additional experiments. I maintain my original score. I recommend discussing methods to identify sweet-spot regions that balance fairness and privacy. For the tabular data, consider presenting the Pareto Front for clarity. Since tabular datasets are not the primary focus, I suggest conducting experiments using fairness metrics like Rawlsian Max-Min fairness, which are more commonly applied to high-dimensional data such as images.

---

> ### Author Response · Authors · 2024-11-27
>
> We really appreciate your additional comment. As per your suggestion, **we added (1) Pareto front results for tabular data (Section E.2) and (2) experimental results with GOLD (Section E.6)**, which supports Rawlsian Max-min fairness and is compatible with PFGuard, as discussed in our previous response. As shown in the table below, employing GOLD achieves the best performance improvement for the smallest group (i.e., worst-case group) – aligning with the goal of Rawlsian Max-min fairness – but GOLD does not necessarily improve group fairness metrics (e.g., KL divergence and Distribution Disparity) or overall utility.
>
> | Method   | Privacy ($\varepsilon$ ) | Fairness (KL ↓)) | Fairness (Dist. Disp. ↓) |  Fairness (Smallest Group FID↓) | Utility (FID ↓) |
> |------------------|-------------|--------------|----------------|-----------|-----------|
> | Privacy-only	       | 10          | 0.177        | 0.383          |  101.39     | **77.97**   |
> | Ours | 10           |  **0.004** |  **0.041**         | 89.43    |   89.76 |
> | Ours + GOLD   | 10           |  0.090      |  0.209       |  **84.52**    | 100.39 |
>
> Additionally, we would like to clarify that **we follow the conventions in the fair generative model literature, which mainly employ group fairness metrics to evaluate fairness in high-dimensional data generation** [Sattigeri et al., 2019; Choi et al., 2020; Yu et al., 2020; Teo et al., 2023]. Rawlsian Max-Min fairness is more commonly addressed in settings without explicit sensitive attributes [Hashimoto et al., 2018; Lahoti et al., 2020; Kenfack et al., 2024], rather than specifically for high-dimensional data. We thus believe the use of group fairness metrics provides more aligned analyses with prior works, while exploring Rawlsian Max-min fairness is indeed an interesting direction.
>
> We thank you again for your insightful feedback, and please let us know if there are any further concerns. We are happy to be engaged with you for further discussions.
>
> Sattigeri et al., "Fairness GAN: Generating datasets with fairness properties using a generative adversarial network.", IBM Journal of Research and Development 2019. \
> Choi et al., “Fair generative modeling via weak supervision.”, ICML 2020. \
> Yu et al., “Inclusive gan: Improving data and minority coverage in generative models.”, ICCV 2020. \
> Teo et al., “Fair generative models via transfer learning.”, AAAI 2023. \
> Hashimoto et al., “Fairness without demographics in repeated loss minimization.”, ICML 2018. \
> Lahoti et al., “Fairness without demographics through adversarially reweighted learning.”, NeurIPS 2020. \
> Kenfack et al., "A Survey on Fairness Without Demographics.", TMLR 2024.

---

> > ### Comment · Reviewer_sxpU · 2024-11-30
> > **Thank you for the responses**
> >
> > Thank you for the additional experiments and your answers to my questions. I have increased my score to reflect the improvement in the paper's quality.

---

> ### Author Response · Authors · 2024-12-01
>
> We truly appreciate you for raising the score. Your feedback was invaluable in improving the quality of our paper, and it was a pleasure to engage in discussions with you.
>
> Warm regards, \
> Authors

---

### Official Review · Reviewer_pijU · 2024-11-11

**Soundness:** 1
**Presentation:** 2
**Contribution:** 1
**Rating:** 6
**Confidence:** 3

**Summary:**

The paper presents PFGuard, a framework for jointly private and fair generative models. The challenges of naively integrating an unfairness mitigation scheme within private generative models are considered. The paper presents an algorithm based on importance sampling to mitigate unfairness first and then privatize using a teacher ensemble a la PATE.

**Strengths:**

- The idea of controlling the fairness-privacy tradeoffs for generative application is worthwhile.

- The writing is clear and easy to follow.

**Weaknesses:**

- **Algorithm likely has unaccounted-for privacy leakage.** The paper claims that the privacy of Private Teacher Ensembles comes from data disjointness alone.  This is not true; that guarantee also depends on random partitioning of the private data; therefore not only disjointness is required but so is random sampling. The importance sampling step that the paper employs cannot, by definition, be completely random as it has to take into account sensitive group. This means that there is an additional privacy cost to this sampling step that the paper simply does not consider. Using importance sampling in DP settings for fairness is not new. In fact, one of aforementioned papers Kulynych et al. 2021 do this while also accounting for the privacy cost of importance sampling.

- **Claims of novelty are exaggerated.** The contribution of the paper is doing private and fair generation; its claim novelty is generation as classification has been done before; yet the private generation is achieved via other methods. In general, reading through Section 4 and especially Section 4.3, the paper is lacking substance regarding the particularities that introducing fairness to private generation brings. Over-relying on prior work like (Jordon et al., 2018; Chen et al., 2020; Long et al., 2021; Wang et al., 2021a) has left the analysis incomplete and without justification.

  For instance, I believe this statement on  Line 315 is wrong:

  > PFGuard preserves any sensitivity as long as the PTEL enforce data disjointness; even with fair sampling, a single data point still affects only one teacher.

	- How come? Can't a point in the minority group be resampled to maintain a similar data distribution across all teachers as assumption (2) (Line 256) requires? Speaking of Assumption 2:

- **Assumption (2)-Line 256 is unrealistic.** paragraph on Line 254 reads:
  > Methodology We now present our sampling technique, which guarantees $B \sim p_{\text {bal }}$ based on SIR. We first make the following reasonable assumptions: 1) each data sample has a uniquely defined sensitive attribute $s \in S$ (e.g., race); 2) $p_{\text {bal }}$ is uniformly distributed over $s ; 3$ ) following Choi et al. (2020), the same relevant input features are shared for each group $s$ between the balanced and biased datasets (e.g., $p_{\text {bal }}(\mathbf{x} \mid \mathbf{s}=s)=p_{\text {bias }}(\mathbf{x} \mid \mathbf{s}=s)$ ), and similarly between the training dataset $D$ and any subset $D_i\left(\right.$ e.g., $\left.p_D(\mathbf{x} \mid \mathbf{s}=s)=p_{D_i}(\mathbf{x} \mid \mathbf{s}=s)\right)$ . We now outline the technique step-by-step below.

	- Assumption 2 here means that samples can be balanced out in terms of sensitive group membership. This assumption is unrealistic. How can the sensitive feature be uniformly distributed; when by definition there exists minority and majority sensitive groups? The only possible way I can think of that would make this work in any practical setting is by resampling minority samples over and over. But that is bound to increase the privacy cost.

- **The paper is missing a number of relevant prior work.** I have already mentioned Kulynych et al. 2021  which essentially does importance sampling for DP-SGD. Also, state-of-the-art for private and fair classification is DP-FERMI by Lowy et al. 2023 is not really considered. Remark 1 on page 4, regarding being the first to revleal that fairness and privacy tecnhiques can counteract each other is not true. Yaghini et al. 2023 and Tran et al 2021  and other make the same observations under PATE classification. I am well-aware of the authors contention in Remark 2 and Section A. So I am going to provide counter-arguments why I cannot accept their line of arguments there in.

	- First, authors claim the other works Jagielski et al. 2019 Mozannar et al. 2020 Tran et al 2021 Lowy et al. 2023 all use DP w.r.t. sensitive attribute hence they account for a different DP definition. While that is true of the first 3, it is not true of Lowy et al. 2023, neither is it true for Kulynych et al. 2021 or Yaghini et al. 2023 who all consider central (i.e. w.r.t all attributes) DP as well. Incidentally, Tran et al 2021 and Yaghini et al. 2023 setting is over PATE which is pretty close to the PTEL setting of the paper modulu the generation part. But as established earlier, the present paper does not advance the generative setting beyond prior work.

	- Second, it is unclear to me why challenges of accounting for the privacy cost of adjusting C plays any role in those works not being considered as baselines. If these methods budget their privacy allocation poorly, doesn't that make for a stark and interesting comparison? To be honest, I do not believe these are the best baselines to compare against but I found this line of argumentation faulty.

**Questions:**

- Can you justify assumption 2 on Line 256? (see my feedback in the weaknesses part)

- Can you include one of the aforementioned baselines?

- Do you acknolwedge the additional privacy cost of importance sampling? Can you address that in a meaningful way?

- Have I misunderstood part of your work? To be clear, I think as is, this paper is not ready for publication. However, I want to be fair and make sure that I have not misunderstood your work. So I'll be happy to engage with you during the rebuttal process.

---

> ### Author Response · Authors · 2024-11-21
>
> To Reviewer pijU (Response 1/2),
>
>
> Thank you for your thoughtful review and constructive feedback. We respond to each of your points below.
> &nbsp;
>
> ---
> > **[W1&Q1]**  Algorithm likely has unaccounted-for privacy leakage. … The importance sampling step that the paper employs cannot, by definition, be completely random … This means that there is an additional privacy cost to this sampling step that the paper simply does not consider.…  Kulynych et al. 2021 do this while also accounting for the privacy cost of importance sampling.
> ---
> We do value your comment, but we would like to clarify **our privacy guarantee is valid**. We newly added a **theoretical proof in our revision** (Sec C.1, highlighted in blue) based on your valuable feedback. Below, we briefly summarize these points.
>
> **The key difference from [Kulynych et al., 2021] lies in “where” importance sampling (IS) is applied.** Their approach directly applies IS to the target model, which requires privacy protection; IS can introduce additional privacy risk to this target model by oversampling certain data points, as you noted. In contrast, PFGuard applies IS to intermediate teacher models, which are not our target models and thus do not require privacy protection [Papernot et al., ICLR 2017; Papernot et al., ICLR 2018; Chen et al., arXiv 2023]. Therefore, we can ensure DP in the target model if we strictly bound the impact of IS (i.e., sensitivity) during the knowledge transfer stage, which learns the target model from the teacher models.
>
> **The use of Private Teacher Ensemble Learning (PTEL) [Papernot et al., ICLR 2017; Papernot et al., ICLR 2018] guarantees the same sensitivity during knowledge transfer regardless of IS-trained teachers.** As explained in Section 4.2, PTEL’s voting scheme bounds sensitivity to “one vote ” because one data sample can affect at most one teacher – due to data disjointness when training teachers – and each teacher can contribute at most one vote. Since IS-trained teachers can still contribute at most one vote, IS does not change the sensitivity of PTEL and thus does not incur additional privacy costs to the target model. We also note that the privacy analysis of PTEL does not rely on random sampling as in DP-SGD, but only on the data disjointness.
>
> We hope this can further clarify how PFGuard avoids IS-related privacy costs, and please let us know if your concern is not fully addressed.
>
> Papernot et al., "Semi-supervised knowledge transfer for deep learning from private training data.", ICLR 2017. \
> Papernot et al., "Scalable private learning with pate.", ICLR 2018. \
> Chen et al., "A unified view of differentially private deep generative modeling.", arXiv 2023.
>
> &nbsp;
>
> ---
> > **[W2]** Claims of novelty are exaggerated. … the paper is lacking substance regarding the particularities that introducing fairness to private generation brings … I believe this statement on Line 315 is wrong.
> ---
> We believe PFGuard’s novelty is to 1) **eliminate such particularities that fairness brings**, which can be a high barrier for users unfamiliar with DP, and 2) **introduce a new framework that can scale to high-dimensional data such as images**, which has not yet been addressed by prior works. In Section 3, we also discuss the challenges of integrating fairness into private generation, including the need to compute additional privacy costs for fairness as noted in your previous comment. PFGuard eliminates such need by preserving the same sensitivity and thus the same privacy analysis regardless of fairness integration.
>
>
> For Line 315, a data sample $x \in D_i$ can be resampled multiple times, **but only used for training one particular teacher $T_i$** that receives a disjoint data partition $D_i$, therefore affecting one teacher and preserving sensitivity as one vote.
>
> &nbsp;
>
> ---
> > **[W3 & Q1]** Assumption (2)-Line 256 is unrealistic 2) pbal is uniformly distributed over s... Assumption 2 here means that samples can be balanced out in terms of sensitive group membership.  How can the sensitive feature be uniformly distributed; when by definition there exists minority and majority sensitive groups? The only possible way … is by resampling minority samples over and over. But that is bound to increase the privacy cost.
> ---
> **$p_{\text{bal}}$ in Assumption 2) represents the ideal target distribution we aim for, not the actual biased training data distribution ($p_{\text{bias}}$ ).** We thus would like to clarify that we do not assume the training data has uniform distribution w.r.t. sensitive attributes. To approximate p_bal given p_bias, PFGuard does resample minority samples multiple times – as you correctly pointed out – but this does not increase the privacy cost as we previously discussed.
>
> &nbsp;

---

> ### Author Response · Authors · 2024-11-21
>
> To Reviewer pijU (Response 2/2),
>
> ---
> > **[W4 & Q2]** The paper is missing a number of relevant prior work. … Remark 1 on page 4, … First, authors claim the other works … it is not true of Lowy et al. 2023 … Tran et al 2021 and Yaghini et al. 2023 setting is over PATE which is pretty close to the PTEL setting of the paper modulu the generation part. But as established earlier, the present paper does not advance the generative setting beyond prior work.
> ---
> We really appreciate your valuable comment. **We corrected the citation error of Lowy et al and removed Remark 1**, where we respect your viewpoint that Remark 1 can appear too bold. **We instead strengthened our discussion in the related work** including the following comparisons:
>
> [Kulynych et al., 2021]
> - [Kulynych et al., 2021] addresses both private and non-private settings, but focuses on fairness in classification accuracy. In contrast, PFGuard focuses exclusively on private settings, covering fairness in both data generation and classification.
>
>
> [Lowy et al., 2023]
> - [Lowy et al., 2023] introduces the first DP fair learning method with convergence guarantees for empirical risk minimization. In contrast, PFGuard provides convergence guarantees for fair generative modeling.
>
>
> [Yaghini et al., 2023] and [Tran et al., 2021]
> - These works rely on public datasets to train student classifiers. In contrast, PFGuard eliminates the need for public datasets by making PATE queries using generated samples from the student generator.
>
>
> We included all the above comparisons in our revision (Section F, highlighted in blue).
>
> &nbsp;
>
> ---
> > **[W4 & Q2]** Second, it is unclear to me why challenges of accounting for the privacy cost of adjusting C plays any role in those works not being considered as baselines … If these methods budget their privacy allocation poorly, doesn't that make for a stark and interesting comparison? Can you include one of the aforementioned baselines?
> ---
> We would like to clarify that **we do include baselines that extend classification methods, such as [Xu et al., 2020] and [Eshipova et al., 2022].** Results in Table 3 show that these methods can incur additional privacy costs due to suboptimal privacy budget allocation, which aligns with your point, and we discuss their behaviors in Section 5.2.
>
> Nevertheless, we do appreciate your feedback on Remark 2 and Sec. A. In our revision, we further clarified: 1) Sec A introduces potential challenges in extending classification methods to generative settings, not claiming this extension is impossible, and 2) we use some possible cases as baselines in our experiments (Sec A, highlighted in blue).
>
> &nbsp;
>
> We again thank you for your constructive feedback, and please let us know if any of your concerns are not fully addressed. We are always happy to be engaged with you for further discussions.

---

> ### Author Response · Authors · 2024-11-25
> **Looking forward to hearing from you**
>
> We understand that this is a busy time for everyone. We would be grateful to know whether our response has addressed your concerns. Please feel free to let us know if you have any remaining questions.
>
> Thank you,
>
> Authors

---

> ### Comment · Reviewer_pijU · 2024-11-29
> **Thank you for the rebuttal.**
>
> Thank you for the detailed rebuttal.
>
> I went through the rebuttal and the updated manuscript. I am convinced by your privacy analysis. Using IS at the level of teachers constitutes a pre-processing step that should not increase the privacy budget. Having said that, in Line 5 of the algorithm you are assuming that you can always subsample a mini-batch from the teacher's data-split that follows the IS ratios. This is not always possible. What if your mini-batch sample has support zero for a particular sensitive subgroup? even if there is support, what if you don't have enough samples to satisfy IS? Do you bootstrap the same samples? If so, that is for sure going to increase the privacy budget.
>
> Also, while the privacy analysis holds; there surely will be a degradation of utility (accuracy) for the teacher model. I don't see any utility argument beyond the empirical results. Can you elaborate?

---

> ### Author Response · Authors · 2024-11-30
>
> We really appreciate your additional response and review of our privacy analysis. We are happy to address your points as follows.
>
> ---
> > Line 5 of the algorithm you are assuming that you can always subsample a mini-batch from the teacher's data-split that follows the IS ratios. This is not always possible. What if your mini-batch sample has support zero for a particular sensitive subgroup?
> ---
>
> Let us first clarify Line 5 of the algorithm and then address your concern about mini-batches with zero support.
>
> [Clarification on Line 5]
>
> The instruction to "Draw a minibatch $\mathcal{B} \subseteq \mathcal{D}_i$ with sampling ratio $w(x)$" means that we *utilize* the importance sampling (IS) ratio $w(x)$ during the sampling process from data subset $\mathcal{D}_i$. It does not assume that the resulting minibatch precisely *follows* a specific distribution. To avoid potential confusion, we will include additional explanation and revise Line 5 as follows:
> - **Original**: Draw a minibatch *with* sampling ratio $w(x)$
> - **Revised**: Draw a minibatch *using* sampling ratio $w(x)$
>
> [Mini-batches with zero support for a sensitive subgroup]
>
> **You are right that mini-batches without any minority samples can occur,** even when using IS to assign higher sampling weights for minority data. For example, if $\mathcal{D}_i$ contains only the majority data samples, any minibatch $\mathcal{B}$ drawn from $\mathcal{D}_i$ will have zero support for the minority data group regardless of IS.
>
> **We thus provide a guideline to mitigate such zero-support scenarios.** Specifically, we limit the maximum number of teachers to probabilistically ensure that each subset D_i includes at least one sample from the smallest subgroup (Sec. 4.2). We also empirically demonstrate how this bound helps avoid fairness compromises arising from these zero-support scenarios (Sec.5.3).
>
> &nbsp;
>
> ---
> > Even if there is support, what if you don't have enough samples to satisfy IS? Do you bootstrap the same samples? If so, that is for sure going to increase the privacy budget.
> ---
>
> You are again right that IS will bootstrap the same samples, resulting in $\mathcal{B}$ with duplicate samples; **however, the use of duplicates does not increase the privacy budget.** The reason is that (1) teacher models – where IS is performed – are trained *non-privately* and (2) duplicate samples affect only one teacher, preserving the sensitivity value as “one vote”, as discussed in W1. Therefore, the privacy analysis remains unchanged even with duplicate samples in $\mathcal{B}$, as mentioned in our revision (Sec. C.1).
>
> We believe that this PFGuard design – achieving fairness through oversampling without breaching privacy – is one of our key contributions, which can effectively address the privacy-fairness conflict. Please let us know if you have any remaining concerns about our privacy analysis.
>
> &nbsp;
>
> ---
> > While the privacy analysis holds; there surely will be a degradation of utility (accuracy) for the teacher model. I don't see any utility argument beyond the empirical results. Can you elaborate?
> ---
>
> As you noted, **our paper provides fairness-utility analyses from conceptual and empirical perspectives, not from a formal perspective.** While formal utility analyses for teacher ensemble structures have been explored in other contexts [Papernot et al., 2018; Bassily et al., 2018; Liu et al., 2021], extending those analyses to our setup poses various challenges due to differences in the problem setting:
> - *Focus on generative tasks*: Generative models often involve more complex architectures (e.g., GANs) compared to prior works focused on classification tasks.
> - *Multiple objectives*: Our approach addresses the interplay of fairness, utility, and privacy, whereas prior works primarily explore utility-privacy tradeoffs.
>
> However, **we believe our conceptual and empirical analyses provide valuable insights, particularly in the underexplored area of integrating fairness and privacy in generative models.** Key conceptual arguments on the utility cost are as follows:
> - Naively integrating fairness in DP generative models can degrade utility due to excessive DP noise, showing the risks of simple sequential designs (Sec. 3).
> - Preserving the main loss function (e.g., adversarial loss) can maintain overall utility despite potential teacher utility loss, demonstrating the advantages of a fair sampling over other fairness methods like additional loss terms (Sec. 4.3).
> - Varying sensitivity values from fairness-privacy interactions during training can lead to unstable utility, introducing new benefits of fairness-privacy decoupling (Sec. A).
>
> We hope these insights on the fairness-utility tradeoff can serve as a first step toward formal analyses, and we fully agree with you that this direction is highly important.
>
> &nbsp;
>
> We again appreciate your additional comment, and please let us know if there are any remaining concerns. We are always happy to engage in further discussions.

---

> ### Author Response · Authors · 2024-11-30
>
> ---
> > References
> ---
> Papernot et al., "Scalable private learning with pate.", ICLR 2018. \
> Bassily et al., "Model-agnostic private learning.", NeurIPS 2018. \
> Liu et al., "Revisiting model-agnostic private learning: Faster rates and active learning.", JMLR 2021.

---

> ### Comment · Reviewer_pijU · 2024-12-02
>
> Thank you for your responses. I think that answers my earlier questions so I'll be raising my score to 6. I have reservations about giving a solid 8 due to concerns about novelty, contribution, and empirical results.
>
> On the empirical side, if you have integrated works such as [Xu et al., 2020] and [Eshipova et al., 2022] (which are non-generative models) then finding a way to integrate the SOTA of non-generative models, DP-FERMI, should be possible. I would not ask you to do that in a rebuttal.

---

> ### Author Response · Authors · 2024-12-03
>
> We truly appreciate you raising the score and sharing your additional comments with us. We would like to address your remaining concerns below.
>
> ---
> > Integrating DP-FERMI [Lowy et al., 2023]
> ---
> We also considered extending [Lowy et al., 2023] as a baseline, but **found their method does not naturally extend to WGANs used in our generative setup**:
> - The computation of ERMI loss [Lowy et al., 2023] requires a model to output *class predictions* (i.e., $\hat{y}$)
> - WGANs do not output class predictions, but output *real-valued scores* to measure the similarity between generated and real samples.
>
> While [Lowy et al., 2023] may extend to other specific generative models such as GANs (i.e., output predictions of real/fake), we (1) opted to use WGANs due to their superior performance over GANs, and (2) instead chose baselines like [Xu et al., 2020] and [Eshipova et al., 2022], which provides more natural extensions to a generative setup, as they are based on DP-SGD, which is also widely used in generative models. We will add more explanations in our current comparison with [Lowy et al., 2023] (Sec. F) based on your valuable feedback.
>
> &nbsp;
>
> ---
> > Novelty and contributions
> ---
> We believe that PFGuard’s novelty and contributions are as follows:
> - **The novelty lies in supporting *modularity* for fairness-privacy based on a new decoupling strategy.** As per your previous concerns, employing oversampling or other fairness techniques on top of privacy techniques can easily lead to privacy breaches. In comparison, PFGuard’s decoupling design supports seamless integration of the sampling technique as well as other fairness techniques (Sec. C.3),  offering flexibility and high modularity for achieving both fairness and privacy.
>
> - **The contribution lies in *scaling* fair and private data generation to high-dimensional data such as images**, bridging the gap between responsible generative models and the high utility of modern models like diffusion models. We believe this alignment with contemporary generative models is both important and valuable.
>
> &nbsp;
>
> We hope our response can address your remaining concerns and again thank you for sharing your valuable comments.

---

### Author Response · Authors · 2024-11-22
**General Response**

We appreciate your thoughtful comments and valuable suggestions, which helped us improve the manuscript. We would like to first address the reviewers’ common question and answer the other comments in each individual response.

&nbsp;

**[Privacy guarantee]**

We would like to clarify that **importance sampling (IS) is independent of the original privacy analysis of Private Teacher Ensemble Learning (PTEL) methods** [Chen et al., 2020; Long et al., 2021; Wang et al., 2021]. IS does not change the data disjointness of PTEL methods and thus does not change the original sensitivity, resulting in the same privacy analysis.

Revision:
- Added Section C.1 to provide a more detailed theoretical proof

&nbsp;

**[Remark 1 on privacy-fairness conflict]**

We really appreciate reviewers’ feedback on Remark 1 – which is seemingly too bold – and thus **removed it in the revision.** We would like to clarify our contributions are (1) demonstrating privacy-fairness conflict in *both directions* using widely adopted data generation techniques, (2) introducing a new design to *fully decouple* privacy-fairness interaction compared to simple sequential designs, and (3) *scaling to image data*, which is not yet addressed by prior works on both private and fair data generation.

Revision:
- Removed Remark 1 to clarify the contributions of the paper
- Further strengthened discussion on related work (Section F)

&nbsp;

**[Privacy-Fairness-Utility Tradeoff]**


**We newly added three additional experiments for more analyses.** We originally used a fixed privacy budget to clearly show privacy-fairness conflict under limited conditions (e.g., number of iterations), but we appreciate great comments to evaluate with different scenarios including varying privacy levels and other data types.

Revision:
- Included visualization of Pareto frontier result to show the privacy-fairness-utility tradeoff of PFGuard (Figure 6, Section E.1)
- Varied privacy levels and compared fairness-utility tradeoff with/without PFGuard (Figure 7 and Figure 8, Section E.1)
- Used tabular dataset to further compare privacy-fairness-utility performances with various baselines (Table 5, Section E.2)

&nbsp;

Chen et al., "Gs-wgan: A gradient-sanitized approach for learning differentially private generators.", NeurIPS 2020. \
Long et al., "G-pate: Scalable differentially private data generator via private aggregation of teacher discriminators.", NeurIPS 2021. \
Wang et al., "Datalens: Scalable privacy preserving training via gradient compression and aggregation.", ACM SIGSAC 2021.

---

### Meta-Review · Area_Chair_wb3N · 2024-12-23

**Metareview:**

The submitted paper introduces "PFGuard," a framework for (image) generation model designed to navigate fairness-privacy-utility trade-offs. Broadly speaking, PFGuard is based on an ensemble of teacher models for balancing fairness and privacy during model training, blending DP mechanisms and balanced mini-batch sampling. Experiments indicate that PFGuard achieves competitive fairness and privacy guarantees without significant utility loss, and show that PFGuard can improve over baseline methods in synthetic data generation and downstream task performance.

Reviewers generally appreciated the focus on simultaneously considering privacy and fairness in generative modeling. They also had mostly a borderline view of the paper. Reviewer sxpU highlighted the experiments on multiple datasets and baselines as a strength, while also raising questions on the paper's chosen fairness definitions. Reviewer WKsH found the approach sound but expressed concerns about the framing of privacy risks,  novelty relative to prior work, and raised issues with the fairness notion used. Reviewer pijU appreciated the modularity of PFGuard’s approach but raised questions regarding the utility impacts of teacher ensemble structures and their scalability. They also expressed serious concerns about positioning relative to prior work. Reviewer oBbq -- the most critical reviewer -- stood by their concerns that the utility of importance sampling is not theoretically justified relative to simpler techniques (e.g., stratified sampling).

Despite these limitations, the reviewers almost unanimously leaned toward a tepid accept. I side with this view, though I would have appreciated if the authors had a more thoughtful discussion on the fairness definitions used in the paper (in line with comments from Reviewre sxpU on connections with other notions of fairness). Since the paper moves the bar forward regarding the interplay of privacy and fairness, I recommend acceptance.

**Additional Comments On Reviewer Discussion:**

During the discussion, reviewer OBbq maintained their main concern that the utility of the importance sampling approach is not theoretically justified relative to simpler techniques like stratified sampling. As this is the main technical innovation of the paper, they noted that a more careful theoretical consideration of the proposed technique is warranted. The other reviewers stood by their more positive view of the submission.

---

### Decision · Program_Chairs · 2025-01-22

Accept (Poster)